# Military Decision-Making Process Enhanced by Image Detection

Nikola Žigulić [1] , Matko Glučina [2] , Ivan Lorencin [2,*] and Dario Matika [3]

1 Faculty of Engineering, University of Rijeka, Vukovarska 58, 51000 Rijeka, Croatia; nzigulic@riteh.hr
2 Department of Engineering, Istrian University of Applied Sciences, Riva 6, 52100 Pula, Croatia; mglucina@iv.hr
3 Varaždin University Center, University North, 31b Jurja Križanića St., 42000 Varaždin, Croatia; dmatika@unin.hr
* Correspondence: ilorencin@iv.hr; Tel.: +385-(0)52-381-412

**Abstract:** This study delves into the vital missions of the armed forces, encompassing the defense of territorial integrity, sovereignty, and support for civil institutions. Commanders grapple with crucial decisions, where accountability underscores the imperative for reliable field intelligence. Harnessing artificial intelligence, specifically, the YOLO version five detection algorithm, ensures a paradigm of efficiency and precision. The presentation of trained models, accompanied by pertinent hyperparameters and dataset specifics derived from public military insignia videos and photos, reveals a nuanced evaluation. Results scrutinized through precision, recall, map@0.5, mAP@0.95, and F1 score metrics, illuminate the supremacy of the model employing Stochastic Gradient Descent at $640 \times 640$ resolution: 0.966, 0.957, 0.979, 0.830, and 0.961. Conversely, the suboptimal performance of the model using the Adam optimizer registers metrics of 0.818, 0.762, 0.785, 0.430, and 0.789. These outcomes underscore the model's potential for military object detection across diverse terrains, with future prospects considering the implementation on unmanned arial vehicles to amplify and deploy the model effectively.

**Keywords:** artificial intelligence; military decision-making process; image detection; intelligence preparation of the battlefield; operation planning; intelligence; imagery intelligence; IMINT; machine learning; deep neural network; You Only Look Once

## 1. Introduction

The armed forces of any nation are a key factor in ensuring the stability of the nation in the current unpredictable environment. The primary goals of any nation's armed forces are based on this, which include protecting the nation's independence, protecting the borders, and maintaining the internal security. These objectives include the following:

- Safeguarding the nation's territorial sovereignty;
- Contributing to regional security systems;
- Supporting civil institutions in the country and abroad;
- Providing overall security for the nation's citizens.

While providing security for the nation's citizens, armed forces have to carry out a wide spectrum of different missions. Force commanders are responsible—each one in his given area of operation, respectively, and on all three levels of command—for accurate and correct mission execution. In order to have the optimal chance of operations success, commanders have to understand the area of operation and given mission, visualize a solution to the problem, and describe the steps of the solution's implementation to his or her subordinates, meaning to issue orders [1,2]. Commanders often face difficult decisions. Moreover, the ever-present uncertainty of war, conflict, and peacetime can make the situation even more challenging. An additional weighting factor are sometimes ethical issues commanders face at all levels of command, as well as soldiers who carry out orders in the field [3–6]. The military headquarters consists of different functional areas and

personnel that participate in a operation planning process known as The Military Decision-Making Process (MDMP). MDMP is a cyclical planning process that involves a unit's commander, headquarter's personnel, subordinate headquarters, and subordinate units. It is used to comprehend a mission and its context, evaluate key components of the mission, create, compare, and evaluate potential courses of action, determine the best course of action for the given mission, generate an operational order, and monitor the mission's execution progress [7]. To make the best decision possible, the commander must have access to relevant, accurate and objective information about his area of operation (terrain and meteorological situation), opposition forces, civilian structures, etc. The commander gets to know such information through the intelligence warfighting function comprising the headquarter's intelligence section and subordinate ISTAR (intelligence, surveillance, target acquisition, and reconnaissance) units [8]. Traditional intelligence collection sources used for intelligence data gathering are as follows:

- Human intelligence (HUMINT) [9];
- Signals intelligence (SIGINT) [10];
- Imagery intelligence (IMINT) [11–13].

While traditional intelligence collection sources have been of the utmost importance throughout the history of warfare, it is really important to highlight that modern intelligence data gathering sources, such as open source intelligence (OSINT) [14,15] and its subcategory, social media intelligence (SOCMINT) [16], show rapid development in the context of the rapid expansion of the cyber warfare domain. Moreover, modern intelligence data collection sources are considerably cheaper to exploit compared to traditional sources.

IMINT, an abbreviation for Imagery Intelligence, is pivotal in contemporary data collection, harnessing information from electro-optical, infrared, RADAR, and LIDAR sensors across a variety of platforms such as land, sea, air, and space platforms. It is important to highlight humans using any type of camera can also gather IMINT data [11]. Unmanned arial vehicles (UAV) stand out for their significant potential in diverse missions within modern warfare and homeland security systems [17]. After data collection, the information is meticulously processed and presented as refined images or videos for end users. The importance of IMINT is underscored by its detailed categorization, as articulated in the NATO doctrine, recognizing 19 distinct categories [18]. Analysts, each specialized in a specific category, process gathered data and create intelligence products. However, challenges arise when gathered data include multiple categories, demanding considerable interpretation time. In scenarios requiring swift processing, the integration of Artificial Intelligence (AI), exemplified by You Only Look Once (YOLO), proves invaluable. This approach allows AI to efficiently process comprehensive data, with analysts subsequently verifying the results. The significance of these data collection methods lies in their capability to provide detailed insights promptly, contributing to timely decision making in dynamic operational environments.

Building upon the information at hand and discerning the data source is achievable through the diverse resources mentioned earlier. Nonetheless, a pivotal inquiry arises concerning the identification of a spectrum of military installations. Bearing this in mind, numerous scholars have undertaken diverse research endeavors in this domain to delve more deeply into various methodologies for detecting adversary objects. The authors of [19] investigated the development of a lightweight military target-detection method, SMCA-$\alpha$-YOLOv5. The method, which involves replacing the focusing module and redesigning the network structure, achieves an exceptional result with an average accuracy of 98.4% and a detection speed of 47.6 FPS. It outperforms competing algorithms such as SSD, and Faster-RCNN, with a significant reduction in parameter cardinality and computational burden. In their research, [20] used the method of Optimal Gabor Filtering and Deep Feature Pyramid Network to utilize a military target-detection dataset named MOD VOC, which was created to meet the PASCAL VOC dataset format standard and includes images primarily sourced from video footage captured by unmanned arial vehicles (UAVs), ground cameras, and internet images. In doing so, they used five artificial intelligence algorithms

(Faster R-CNN, DSOD300, DSSD513, YOLOv2 544, and their filtering method) to compare the individual performance of the best model for each algorithm. The best results for MOD VOC in terms of accuracy, recall, and average fps were achieved by their model in the amount of 88.76%, 78.45%, and 30.35, respectively, while the worst results were achieved by DSSD513 in the amount of 73.57%, 64.56%, and 42.43. Kong et al. [21] used a military target dataset showing armed individuals with different weapons to improve the detection performance of the proposed YOLO-G algorithm. The authors introduced improvements compared to the YOLOv3 framework, including a lightweight GhostNet for improved accuracy and speed in detecting military targets. The dataset evaluation showed a 2.9% improvement in mAP and a 25.9 FPS increase in the detection rate compared to the original YOLOv3, highlighting the effectiveness of their improved algorithm. Wang and Han [22] introduces the YOLO-M algorithm for military equipment target recognition, addressing challenges in small target detection. By incorporating the C3CMix module and modifying the activation function in YOLOv5, the proposed algorithm maintains high accuracy while reducing parameters, resulting in a 95.2% average accuracy, an 18.8% reduction in parameters, and a 14.5% decrease in computation. These improvements make YOLO-M well-suited for deployment in military equipment target recognition applications. Du et al. [23] investigated military vehicle object detection based on hierarchical feature representation and refined localization for the detection of military objects in the desert, grass, snow, city, and others. The authors applied R-FCNN, SSD, YOLOv3, YOLOv4, Faster R-CNN, and MVODM, i.e., a novel algorithm created by the author. The models were trained on three different types of test datasets, i.e., large-scale, small-scale, and all subset test datasets. The best results were shown by MVODM (novel algorithm) for a large-scale dataset with evaluation metrics resulting in the amount of 85.6% mAP, while the worst performance was YOLOv3 for a small subset of data in the amount of 54.9% mAP. Nelson and McDonald [24] developed the Multisensor Towed Array System (MTADS), demonstrating its effectiveness in detecting buried unexploded ordnance with an outstanding probability of detection (0.95 or better). The system results highlight its precision in locating ordnance at self-penetrating depths, providing a cost-effective and accelerated approach to remediation compared to standard techniques. Pham and Polasek [19] address the challenges of surface object detection in urban environments, utilizing both the infrared and visible spectra. The paper aims to develop an algorithm for detecting and selecting objects of interest, particularly civil automobiles resembling military equipment, captured by infrared and visible cameras in various outdoor conditions. The proposed algorithm involves determining optimal threshold values for image conversion in changing environmental conditions and colors, tested on static images, and extended to dynamic object detection, selection, and tracking through video processing. Additionally, the study emphasizes the use of threshold adjustment techniques to optimize object detection. The authors of [25] introduce a method inspired by EfficientDet trackers for classifying maritime military targets in high-resolution optical remote sensing images. The approach involves constructing a multilayer feature extraction network with attention mechanisms, utilizing ReLU activation, and employing deep feature fusion networks and prediction networks to accurately identify various types of military ships. The trained model was tested on six classes of vessels, achieving the best detection performance in the GW class with precision and recall values of 0.983 and 0.945, respectively. The lowest detection results were observed in the SS class, with precision and recall values of 0.974 and 0.822, respectively.

Considering intelligence data collection from IMINT sources, special attention in this paper is given to collection with unmanned arial systems (UAS) and challenges that emerge while collecting and analyzing data collected with UAS. Current NATO UAS classification recognizes three different UAS classes [26]. Those are classes I, II, and III. The main class feature that these classes are differentiated by is their maximum takeoff weight (MTOW). Class I includes UAS whose MTOW does not exceed 150 kg. Class II includes UAS whose MTOW is more than 150 kg, but less than 600 kg. Lastly, class III includes UAS whose MTOW exceeds 600 kg. A bigger MTOW means the UAS is capable of carrying larger and

better quality payload and is therefore able to execute missions which are more complex. Moreover, the mentioned classes are also differentiated by the level of command they are subordinated to, their operating altitude, their range, and their operating range. It is important to highlight that the flying crew of classes II and III consist of two crew members. The first one is a pilot who is responsible for the execution of flying operations of the unmanned arial vehicle (UAV), while the second member is a payload operator, responsible for gathering IMINT data. On the other hand, UAS class I systems are operated by only one operator, who is a pilot and also a payload operator. This means that class I operators are exposed to a rather high mental workload in terms of multitasking (operating the sensor and watching all flight parameters at the same time) during the whole mission, which may last up to 6 h. This high mental workload results in a lot of accumulated fatigue, meaning that the UAS operator experiences lower concentration in the later stages of a mission. In order to make UAS class I operators' work easier, using algorithms with a small detection time, such as YOLOv5, would be highly beneficial. It would detect objects of interest instead of the operator and he would only confirm, with a bigger zoom, if that is really the object of interest or he can move on. Moreover, the operator could concentrate more on flying parameters and noticing any malfunction. Furthermore, another issue that may emerge is that unit's headquarters and all subordinate units do not have enough capacity to analyze all gathered IMINT data or just do not have enough time to do it. Current practice for IMINT data analysis is data being analyzed by an IMINT analyst who is narrowly specialized in one of 19 different IMINT categories according to NATO doctrine. Sometimes IMINT data may consist of different elements which belong to different categories. This may pose a problem during complex operations which require a fast flow of information, especially if a country's armed forces do not have an IMINT analyst specialized in each class and one analyst has to analyze a few different classes. YOLOv5 is an ideal solution for this problem due to its speed, reliability, and accuracy. It is capable of handling various challenges in the detection, classification and segmentation of objects.

According to the presented literature overview and problem description, the following questions can be asked:

- Is it possible to create a military dataset by using publicly available data?
- Is it possible to use object detection algorithms such as YOLOv5 for military object detection?
- How does the proposed method simplify and contribute to improving the quality of military decision-making?

The aim of this article is not to showcase the methodology of UAV control, but rather to demonstrate how a dataset with military applications can be compiled using publicly available data, which can be utilized for the development of detection software. For this reason, the paper does not focus on the construction elements of drones or the methodology of their flight control.

## 2. Materials and Methods

This subsection describes the method of collecting the dataset, which classes it consists of, and how many images the total dataset contains. Furthermore, the methodology of the You Only Look Once fifth generation detection algorithm (YOLOv5) will be presented together with the associated parameters and the principle of convolution neural network (CNN) training.

### 2.1. Materials

The dataset used in this work was obtained through open-source websites. The websites included were as follows: various official countries' armies' websites, different countries' official army profiles on popular video streaming platforms, open-access gore websites, and news portals. The used dataset represents various military equipment and weapon systems used in modern combat. These weapon systems have different countries of origin and were recorded from different angles and in different conditions. Some weapon

systems were recorded from the air by UAV, while others were recorded from the ground by humans. Moreover, the dataset has examples of maneuver forces' vehicles such as tanks, infantry fighting vehicles, and armored personnel carriers, examples of air force vehicles such as transport or assault helicopters and transport or assault airplanes, and examples of engineering vehicles, anti-aircraft vehicles, and artillery vehicles. This research's focus is weapon systems only, and therefore people were not included. Furthermore, a large portion of the collected videos are real combat footage, recorded mostly in the current conflict between the Russian Federation and Ukraine. Other videos were taken at various military exercises or expos and posted on the internet. Each video obtained through open source websites was annotated using the Dark label annotation tool [27]. The dark label tool is utility software that can label and name the object bounding boxes in videos and photos. Additionally, it may be used to mosaic image regions, sample moving images, and crop videos. Handling this software is quite simple. First, the class names are defined in the darklabel.yml file. After that, the YOLO annotation format is selected, and the process of labeling images of the desired classes begins. The classes of interest for this research are as follows:

1. Tank (TANK);
2. Infantry fighting vehicle (IFV);
3. Armored personnel carrier (APC);
4. Engineering vehicle (EV);
5. Assault helicopter (AH);
6. Assault airplane (AAP);
7. Transport airplane (TA);
8. Anti aircraft vehicle (AA);
9. Towed artillery (TART).

It can be seen that the dataset consists of 9 classes, and after labeling all videos (that is, images that were broken into frames by the Dark label software), the total dataset consists of 24,178 images. The distribution of data, i.e., the number of annotated images, can be seen in the histogram shown in Figure 1. The largest group of annotated images is for class AA, while the smallest group is for class EV. It is obvious that the classes are not uniformly distributed, which represents a challenge in the precise detection and classification of individual classes.

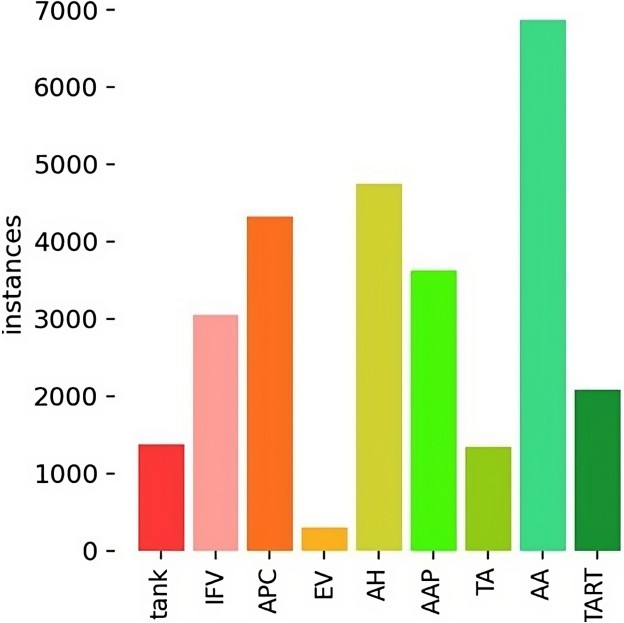

**Figure 1.** Histogram of all used classes in this research.

The dataset is divided into train/test/validation sets in the ratio of 70/20/10, respectively. As a result, 16,900 images form the training set, 4849 the test set, and 2429 the validation set.

Each image used in the training of the YOLOv5 detection model carries with it important information that shapes the neural networks within the model. Each image pixel has a functional value and contributes to the learning process, creating a deeply connected network of neural connections. Accordingly, the presentation of these images in the paper is of essential importance, as it enables the reader to see the specific characteristics and complexity of the input data that influence the final results of object detection. Several examples can be seen in Figure 2.

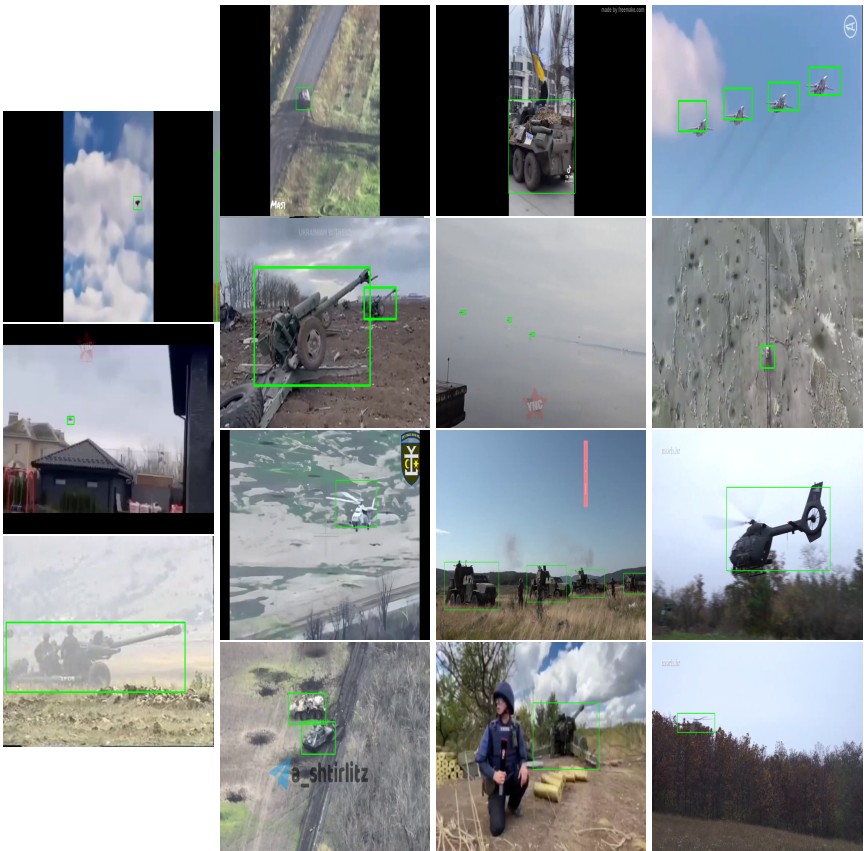

**Figure 2.** Visual representation of images employed in YOLOv5 algorithm training.

In addition, the image display includes a visual analysis that reveals the complexity of military nature and additional challenges in training a detection algorithm like YOLOv5. The pictures clearly show the shades of camouflage used to hide objects from enemy observation. This characteristic makes training a detection model more challenging, requiring precision in object recognition even under difficult-to-see conditions. Training an algorithm to accurately detect military objects despite different forms of camouflage is an important goal in the development of reliable detection algorithms for military applications. The images shown, labeled as in Figure 2, provide concrete examples of these challenges and illustrate the need for sophisticated detection algorithms.

Military Data Curation Process: Unveiling the Significance and Challenges of Comprehensive Datasets in a Strategic Context

The quality of the dataset is a key prerequisite for successful research in the field of artificial intelligence, and this especially applies to the development and training of algorithms. According to generally accepted standards, as much as 70% of the total effort in the process of training an AI algorithm is devoted to the collection, processing,

and preparation of data. In order for research to achieve an acceptable level of precision and reliability, it is essential to have a high-quality dataset.

Figure 3 illustrates the arduous nature of the dataset collection process, comprising a series of sub-steps that demand considerable time investment. The compilation of a dataset to a level acceptable for annotation, let alone training AI algorithms, is a painstaking endeavor. This visual representation underscores the pivotal role of a well-curated dataset in the development of artificial intelligence. It highlights that research based on vaguely defined or low-quality data may yield unrealistic results and draw incorrect conclusions. Various methods, including surveys, camera recordings across different devices, and even oral transmission of information, can be employed in data collection. Diverse collection conditions, encompassing factors such as the environment, weather conditions, and context, can exert a substantial impact on research outcomes. Hence, it becomes crucial to meticulously account for these factors when curating a dataset.

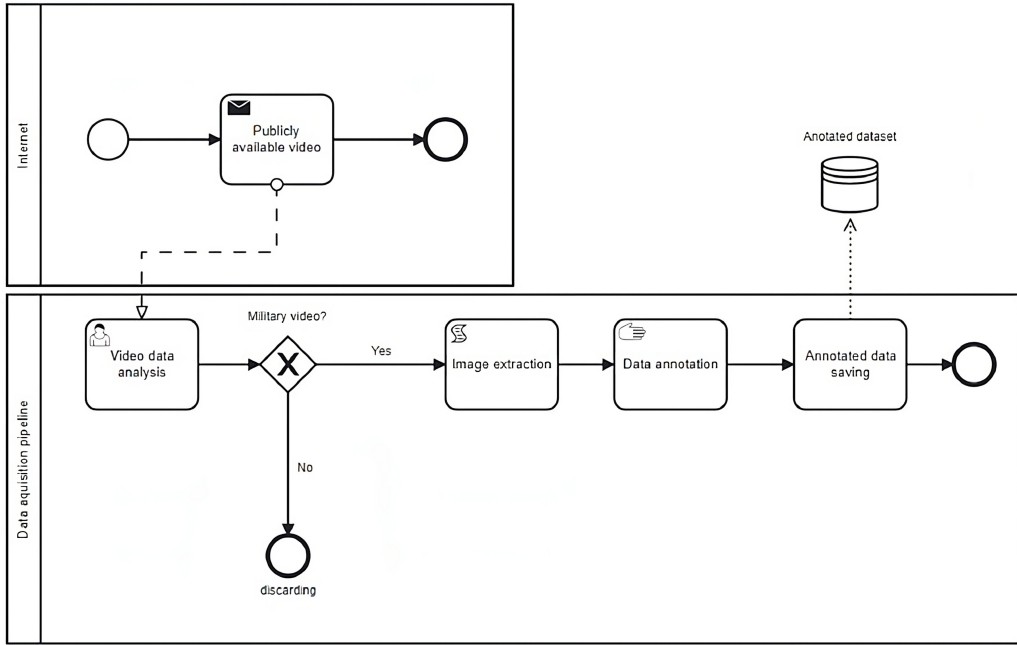

**Figure 3.** Flowchart utilized in this research for military data collection protocol.

As depicted in Figure 3, the data acquisition process comprises two pivotal phases. The initial phase entails a comprehensive exploration of available videos and diverse image repositories accessible on the internet. The primary objective of this stage was to meticulously curate a set of high-caliber computer data that would undergo subsequent processing. The quest for data traversed various platforms, including YouTube, Google Images, Wikimedia Commons, and others.

Prior to commencing the analytical phase, meticulous consideration was devoted to adapting and scrutinizing all designated video formats. This involved the critical assessment of video quality, addressing challenges related to perspective constraints, accounting for temporal variables, and discerning potential manipulations and edits within the material. Subsequent to the successful compilation of a substantial dataset, comprising 101 videos averaging 180 s in length, each video comprising 60 frames per second (FPS) for a cumulative total of 1,090,800 images, a judicious analysis and evaluation process ensued. Each image underwent scrutiny as a prospective candidate for annotation, necessitating precise labeling and identification of pertinent information to ensure a nuanced and pertinent analysis in subsequent research endeavors. Following the implementation of solutions to potential challenges, a comprehensive review process was undertaken to assess all acquired frames. The primary objective was to ascertain the presence of objects exhibiting military characteristics, thereby enhancing the overall quality of the dataset and

refining the definition of the target class of military objects. Should a specific frame fail to meet the pre-established criteria, signifying a departure from the defined conditions, it was systematically rejected from further consideration. This meticulous curation process ensured that only frames aligning with the specified criteria were retained, contributing to the precision and reliability of subsequent analyses and findings within the research context. Following the successful resolution of potential challenges, a meticulous examination of all acquired frames was initiated, with the objective of discerning objects exhibiting military characteristics. The primary objective of this procedure was to enhance the quality of the dataset and provide a more accurate delineation of the presence of the targeted class of military objects. Instances where the specified conditions were not met, where a particular frame failed to meet the predefined criteria, prompted its systematic exclusion. This systematic curation process was undertaken to uphold the consistency and high quality of the data utilized in the course of the research.

Upon the successful extraction and meticulous curation of images, the subsequent step involved annotating the images to facilitate the training of the YOLOv5 detection algorithm. In this phase, dual considerations were paramount. Firstly, the military perspective was taken into account, encompassing elements deemed significant from the standpoint of a soldier, lieutenant, and the like. Simultaneously, the YOLOv5 algorithm was loaded and subjected to rigorous testing to evaluate its performance under real-world conditions. A more intricate exposition of the data collection process, along with illustrative examples, is presented in Figures 4–9.

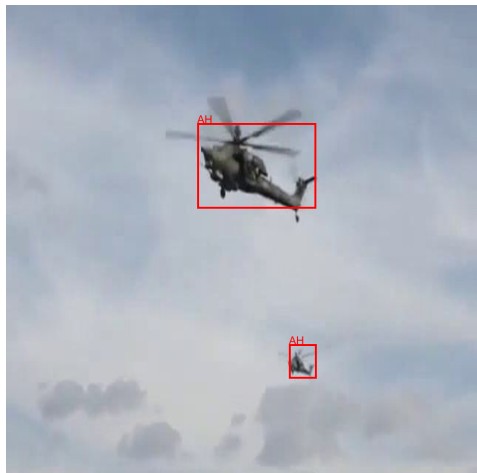

**Figure 4.** Example dataset featuring a military object target class AH in plain sight.

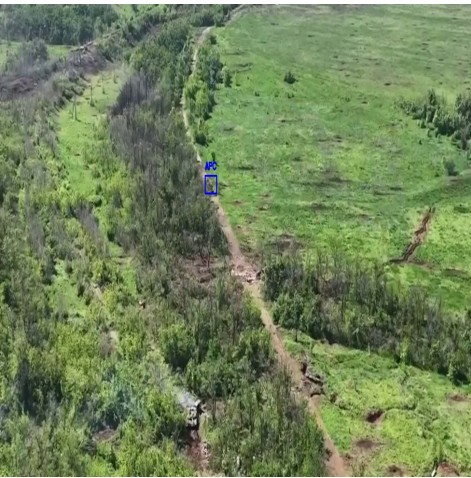

**Figure 5.** Example of an APC class in dense forest terrain.

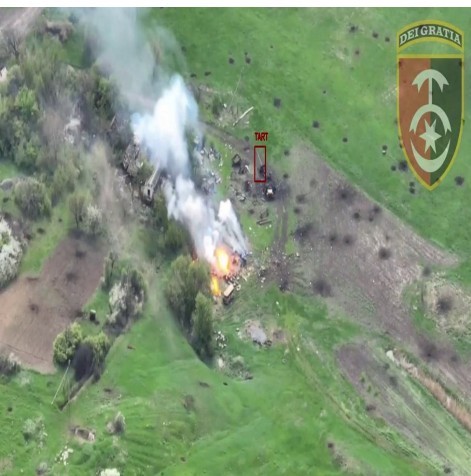

**Figure 6.** TART class blending into battle focus, obscured by smoke and local terrain.

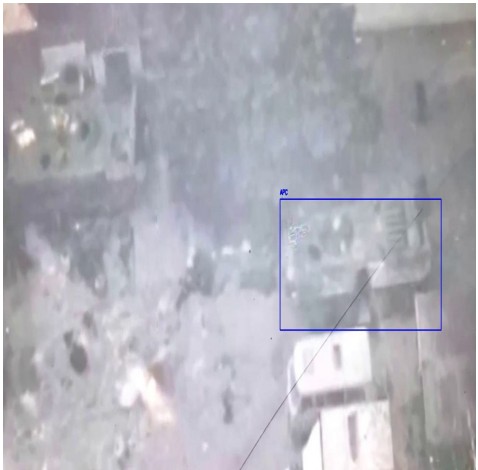

**Figure 7.** Example of an APC class in an urban setting blending with local buildings.

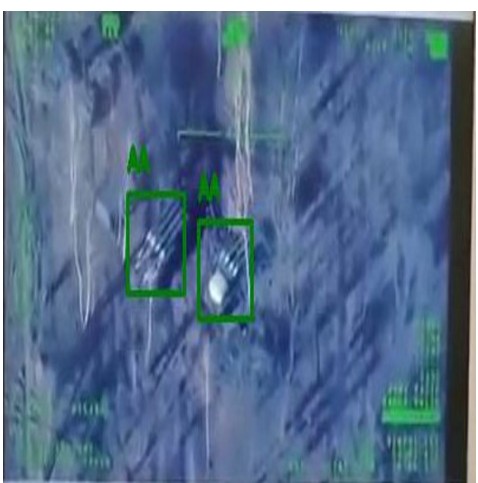

**Figure 8.** Example of AA class merging with the surroundings.

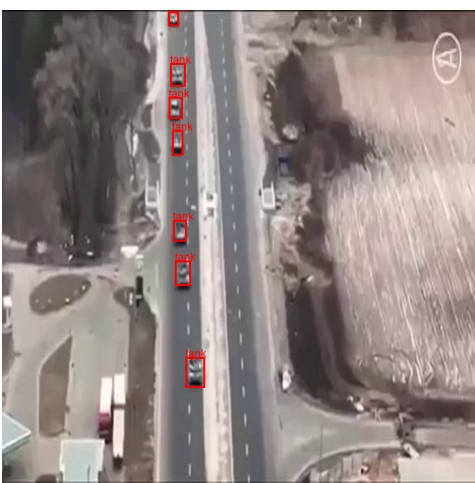

**Figure 9.** Example of a tank convoy moving towards a destination.

From military operations planners' and military commanders' standpoint, the significance of this dataset lies in the variety of military equipment classes included. An algorithm trained with this particular dataset can help military commanders and military headquarters to have better situational awareness about opposition forces' structure and operational capabilities, especially in intense battle rhythm operations, when quick information flow is essential. From a machine learning standpoint, the significance of this dataset lies in its pivotal role in enhancing the robustness and generalization of algorithms. The incorporation of luminance contrast diversifies the learning experience, enabling the algorithm to adeptly discern varying levels of luminance and thus bolstering its resilience to changes in lighting conditions. Furthermore, the dataset richness in colors and textures contributes to the capacity of the algorithm to generalize across diverse object types and backgrounds, as exemplified in Figures 7 and 8. In terms of preventing overfitting, the dataset inclusion of different perspectives is instrumental in averting model specialization to particular positions or viewing angles. Moreover, the incorporation of varied recording conditions, such as distinct cameras and weather scenarios, serves as a safeguard against overfitting to a specific dataset, as demonstrated in Figures 5 and 7. The dataset emphasis on increasing variation is evident through its incorporation of geographical and environmental diversity. This inclusion exposes the algorithm to different locations and environments, fostering an ability to adapt to various conditions, such as multiple entry with similar properties as shown on Figure 9. Notably, the dataset captures scenarios where autonomous humans (AH) are situated outdoors, or within dense vegetation such as tall trees, or the elevated roofs of buildings. The dataset further contributes to the model's versatility in solving various problems. The introduction of different object sizes and distances enables the model to develop proficiency in accurately detecting objects within diverse contexts. This is illustrated, for instance, in Figure 4, where AHs exhibit scaling, with one AH appearing smaller in relation to the other. Additionally, the dataset encompasses variations in object positions within images, facilitating the model's ability to recognize objects across different parts of an image. Lastly, the inclusion of diverse time periods in the dataset, encompassing night, day, snow, dust, smoke, and more, augments the model's performance on real-world data as in Figure 6. This is particularly relevant in the context of autonomous vehicles, where variations in driving conditions, including night, day, rain, and snow, contribute to preparing the model for a spectrum of real road situations.

As perceived from the perspective of ML engineers and AI algorithms, the presented dataset furnishes exemplary instances under diverse conditions. This dataset serves as a comprehensive evaluation ground, testing the algorithm's performance capabilities in varied scenarios. Additionally, it caters to the specific needs of military operators, providing valuable insights tailored to their operational requirements.

The description of the dataset is finished with this part, and the detection algorithm used to solve the given problem is presented in the next subsection.

### 2.2. Methodology

YOLOv5 is an ideal solution for this problem due to its speed, reliability, and accuracy. It is capable of handling various challenges in the detection, classification, and segmentation of objects [28,29]. This single-stage object detector consists of a backbone, neck, and dense prediction (head) as shown in Figure 10. The algorithm accelerates the process by analyzing the data and the analyst only needs to verify the results.

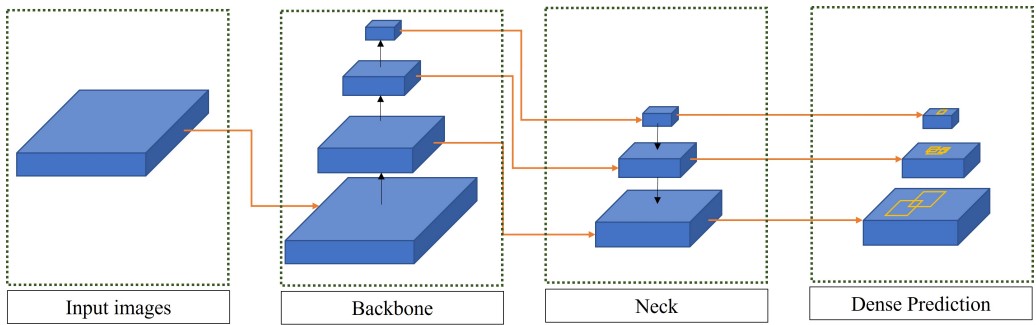

**Figure 10.** Single-stage (YOLO) detector architecture.

We opted to use YOLOv5 for the detection of military objects within a dataset we independently gathered. Our decision to utilize YOLOv5 over newer methods stemmed from the intention to showcase our capability to detect military objects within a publicly available dataset. YOLOv5 was chosen as it represents a stable and proven methodology for object detection tasks. Given its established track record and reliability, employing YOLOv5 allowed us to demonstrate our proficiency in identifying military objects reliably and accurately. While newer methodologies might offer enhancements or improvements, opting for YOLOv5 allowed us to focus on demonstrating our proficiency within a well-established and widely recognized framework. It ensured that our results could be easily reproducible and comparable against existing benchmarks in the field of object detection, reinforcing the reliability and stability of our approach. The YOLOv5 detection algorithm excels in achieving an optimal balance between speed and accuracy, featuring a refined architecture suitable for implementation on resource-constrained microcontrollers [30]. From the compact YOLOv5 nano model to larger variants, these models exhibit memory weights conducive to diverse applications, including military contexts. Comparative evaluations against DETR and EfficientDet underscore YOLOv5's superiority, particularly in challenges like crop circle detection, where it outperforms with a recall of 0.98, surpassing DETR and EfficientDet 0.68 and 0.85, respectively, alongside precision values of 0.77 and 0.91 [30–32]. In scenarios involving overlapping object detection in kitchens, YOLOv5 excels by producing accurate frames and demonstrates superior performance compared to Faster R-CNN [33]. The study reveals YOLOv5's accuracy of 0.8912 (89.12%) outshining that of Faster R-CNNs of 0.8392 (83.92%), underscoring its effectiveness in handling complex scenarios. Beyond object detection, YOLOv5 exhibits computational efficiency, outpacing sophisticated methods like RetinaNet, as observed in Liu et al. [34–36]. Despite not universally achieving optimal results, the model performance nuances are vital considerations in addressing computational complexities during training. The versatility of YOLOv5 extends across domains, including the analysis of secondary waste treatment processes, applications in autonomous vehicles, and weed growth detection. Particularly noteworthy is its exceptional interference reduction and efficiency in weed growth detection, positioning YOLOv5 as an optimal choice for military applications, as emphasized by Almalky et al. [37]. Owing to foundational principles and analogous use cases, YOLOv5, characterized by its intricate, comprehensive, precise, and rapid attributes, undergoes scrutiny for the detection of military objects.

The input of the algorithm contains the images from which the dataset is composed: in this case, any image from a possible 24,178 in total. The backbone is a pre-trained network whose role is the extraction of features from images. Then, using the backbone, the spatial resolution of the image is reduced, and the resolution of the features (channel) is increased [38]. The role of the neck is a pyramid of features, that is, to extract feature pyramids. The neck [39] has a generalization value which helps the model to generalize different objects with multiple sizes and scales. At the end of the single-stage detection algorithm, there is a head that is used to perform the final actions in which anchor boxes are applied to feature maps and the output results of the algorithm are rendered as object classes, bounding boxes, and object scores, i.e., the percentage of certainty that it is that object. YOLOv5 has developed five models of different sizes: nano, small, medium, large, and extra large models. As far as model operations are concerned, there is no difference, that is, the principle of operation is the same for all, but the difference occurs in terms of layers and parameters (later on, in the inference speed and memory size of the model) [28]. As previously defined, the single-stage detector, including YOLOv5, consists of three components: the first is the backbone, which in this case is CSP-Darknet53, the neck, i.e., SPP and PANet in the case of YOLOv5, and the head, which is identical to the previous generation of the YOLO algorithm (YOLOv4 [40]). The listed components are shown in Figure 11.

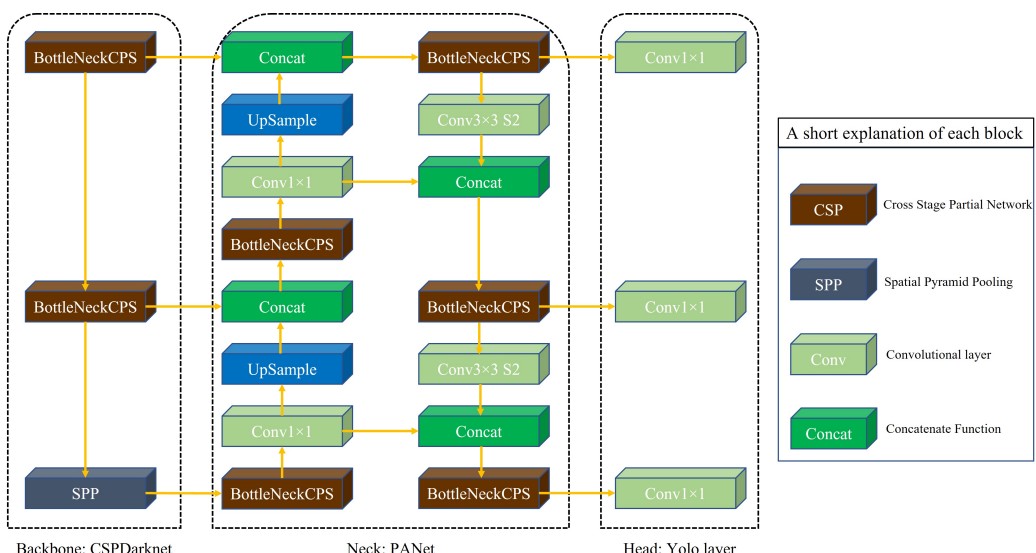

**Figure 11.** The YOLOv5 network architecture.

YOLOv5 is an update of previous versions of the YOLO detection algorithm. In the fifth version, CSP-Darknet53 is used, which is actually CNN Darknet53 [38], which was used in YOLO version 3 (YOLOv3) [41], but the authors of YOLOv5 improved the algorithm by implementing the CSP network strategy. YOLO is a deep neural network (DNN) that utilizes residual and dense blocks to ensure the flow of data into the deepest layers of the neural network (NN) and prevents the problem of vanishing gradient. The problem of vanishing gradients in machine learning is encountered when using gradient-based learning techniques and backpropagation to train artificial neural networks. This is because each neural network weight is adjusted in accordance with the partial derivative of the error function with respect to the current weight in each training iteration [42]. The CSPNet approach employed by YOLOv5 has the benefit of reducing the number of parameters and computations, or FLOPS, and increasing the speed of inference. This is an important factor to achieve real-time object detection. The method divides the feature map of the base layer into two sections and then links them through the cross-stage hierarchy, which helps to address the issue of redundant gradients.

The neck in YOLOv5 brings two big changes. The first variant is Spatial Pyramid Pooling (SPP), while the second is the Path Aggregation Network (PANet) which is integrated using BottleNeckCSP in its architecture which is described below.

- PANet is a feature pyramid network that was used in the predecessor of the YOLOv5 version (in the fourth generation) to improve information flow and contribute to pixel localization. In the YOlOv5 version, the network has been improved by integrating the CSPNet strategy.
- SPP bulk is intended for the accumulation of information received from the input and returns an output of a fixed length. This increases the influence of the receptive field and separates the most important and relevant features without reducing the speed of the network.

The YOLOv5 head is configured identically as in the cases of the detection algorithms YOLOv3 [41] and YOLOv4 [40]. It consists of three convolutional layers that find and predict the location of the bounding box, the confidence coefficient, and the object class.

YOLOv5 Configuration

Table 1 displays the hyperparameters that were altered during the training of the YOLOv5 algorithm. Each resulting model was trained over 500 epochs, using different image resolutions and optimizers. Starting with a resolution of $512 \times 512$ pixels, three models were trained, differing in the optimizer used—Adam, AdamW, and the SGD optimizer. After these three models were trained, the training process became more complex, increasing the image resolution from $512 \times 512$ to $640 \times 640$. Likewise, these models were also trained with the aforementioned optimizers. Finally, the highest image resolution in this study was $1024 \times 1024$ pixels, with all three optimizers used for training for 500 epochs. The common element of all obtained models is the Patience parameter, which limits unnecessarily prolonged training. In other words, if the results do not improve significantly after 100 epochs, the training is stopped and the last obtained epoch is taken as the final training result.

**Table 1.** YOLOv5 hyperpatameter configuration.

| | | | Hyperparameters | |
|---|---|---|---|---|
| **No** | **Image Size (in Pixels)** | **Epochs** | **Optimizer** | **Patience** |
| 1 | | | Adam | |
| 2 | 1024 | | AdamW | |
| 3 | | | SGD | |
| 4 | | | Adam | |
| 5 | 640 | 500 | AdamW | 100 |
| 6 | | | SGD | |
| 7 | | | Adam | |
| 8 | 512 | | AdamW | |
| 9 | | | SGD | |

The Adam optimizer is one of the most common choices for computer vision tasks [43]. The optimizer is designed to be suitable for non-stationary tasks and problems with a lot of influence of noise or sparse gradients [44]. Using the Adam optimizer, the weight values are updated according to the following mathematical expressions [44]:

$$w_t = w_{t-1} - \eta \frac{\hat{m}_t}{\sqrt{\hat{v}_t} + \epsilon} \tag{1}$$

where $\hat{m}_t$ is defined as

$$\hat{m}_t = \frac{m_t}{1 - \beta_1^t}, \tag{2}$$

where $\hat{v}_t$ is defined as

$$\hat{v}_t = \frac{v_t}{1 - \beta_2^t}, \tag{3}$$

$m_t$ can be defined as the current average of the gradients and can be described as

$$m_t = \beta_1 m_{t-1} + (1 - \beta_1)G, \tag{4}$$

Finally, $v_t$ can be defined as average of squared gradients with following equation:

$$v_t = \beta_2 v_{t-1} + (1 - \beta_2)G^2 \tag{5}$$

with $G$ mathematically described as

$$G = \nabla_w C(w_t) \tag{6}$$

Equations (1)–(6) contain the following:

- $\eta$ represents step size or learning rate.
- $\epsilon$ is a correction parameter, i.e., a number of the order of $10^{-8}$ and smaller that prevents the possibility of diverging results.
- $\beta_1$ and $\beta_2$ are forgetting parameters; the running average is updated more quickly when either one is lower (and hence the faster previous gradients are forgotten).
- $C(w_t)$ is a cost function.

By separating the weight from the gradient update, AdamW is a method-based stochastic optimizer that controls a common implementation of weight decay. The regularization parameter $L_2$ in the Adam optimizer is usually implemented using Equation (7) where $\omega_t$ is the rate of the weight decay in time $t$ [45]:

$$g_t = \nabla f(\theta_t) + w_t \theta_t \tag{7}$$

while the difference between Adam and AdamW is the adjustment of the weight decay parameter in the gradient update [45]:

$$\theta_{t+1,i} = \theta_{t,i} - \eta \left( \frac{1}{\sqrt{\hat{v}_t + \epsilon}} \cdot \hat{m}_t + w_{t,i} \theta_{t,i} \right), \forall t \tag{8}$$

SGD [46] is one of the simplest optimizers for adjusting weight values and is mathematically described as

$$w_{t+1} = w_t - \alpha \cdot g_t, \tag{9}$$

where the following applies:

- $w$ is the previous value of the weight, i.e., parameter;
- $g$ modifies the gradient of the model;
- $t$ is a time step;
- $\alpha$ is a global learning rate for the given optimizer.

In SGD, the optimizer predicts, based on a mini-batch, the direction of the steepest fall and moves in that direction. Due to the limited step size, SGD can easily become stranded in peaks or local minima. The gradient descent optimization technique can be improved by adding momentum, which allows the search to gain momentum in a certain direction in the search space, avoid noisy gradient oscillations, and move across flat areas of the search space. The momentum parameter can be expressed as

$$v_{t+1} = \beta \cdot v_t + g_t. \tag{10}$$

By adding the momentum parameter from Equation (10), the ultimate mathematical form of the SGD optimizer is formed:

$$w_{t+1} = w_t - \alpha \cdot v_{t+1}. \tag{11}$$

$\beta$ represents the weightage that is going to assign to the past values of the gradients and its value is lower than 1, i.e., SGD accelerates in the directions of constant descent, and this acceleration makes it possible to avoid the plateau and makes the model less sensitive (in local minima areas).

Each of the models obtained in this investigation was trained on a small NN YOLO v5 at the intersection over union (IoU) threshold in the amount of 0.6. IoU represents the ratio of the intersection area (overlap marked with green) and the union of the training (predicted and marked with yellow) bounding box and the ground truth bounding box shown in Figure 12a. In a real environment, due to multiple factors, the algorithm cannot always 100% predict the boundary box. So, for example, the overlap in Figure 12b is not 100% but rather lower, while the area of the union is complete, that is, 100%. Thus, in the example of Figure 12b, if the intersection area is 0.6, i.e., 60%, and the union area is 1, i.e., 100%, then the IoU value is 0.6.

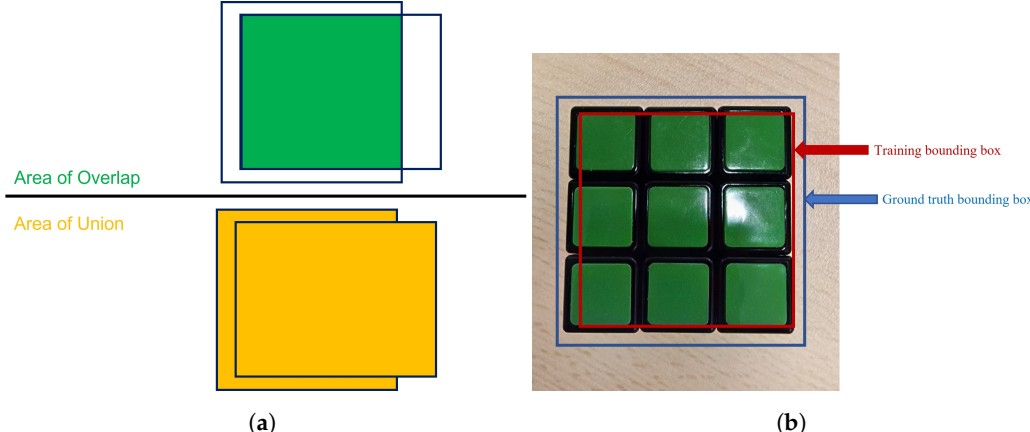

**Figure 12.** Graphical and realistic representation of training (predicted) bounding box and ground truth bounding box together with a graphical representation of IoU. (**a**) Graphic representation of the IoU. (**b**) Presentation of training (predicted) bounding box and ground truth bounding box.

## 3. Results

In this section, the results obtained by the YOLOv5 detection algorithm are presented. Each model is presented with the appropriate parameters, evaluation metrics are also defined, which evaluate the performance of each of the obtained models. In the end, there is a discussion about the performance of the model, which model performed better and which performed worse, and what could potentially be the cause of it.

*Obtained Results of the Trained Models*

The results obtained from this research are presented in Table 2 below. Precision, recall, map@0.5, mAP@0.95, and F1 score are shown for each model. After a short presentation of the results, the confusion matrix for the best model for each resolution will be described.

Table 2 shows the results obtained in this research. It is of great importance to first explain the difference between the two evaluation metrics mAP, i.e., mAP@0.5, and mAP@0.95. The main difference between mAP at a value of 0.5 (mAP@0.5) and 0.95 (mAP@0.95) lies in the detection precision requirements:

- mAP@0.5 represents the average detection accuracy when using a threshold of 0.5 for successful object recognition. This means that an object is considered correctly detected if it overlaps with the reference object (ground truth) by at least 50%.

- mAP@0.95 represents the average detection accuracy when using a high threshold of 0.95 for successful object recognition. This means that an object is considered correctly detected only if it overlaps the reference object by 95% or more.

Basically, mAP@0.95 represents a stricter criterion than mAP@0.5, requiring greater precision in object detection. This implies that mAP@0.95 will be lower because achieving a high degree of overlap of 95% with reference objects is a challenge. Therefore, there is a significant discrepancy in the values between these two measurement metrics.

**Table 2.** The training outcomes achieved through machine learning techniques.

| Optimizer | Precision | Recall | mAP@0.5 | mAP@0.95 | F1 Score |
|---|---|---|---|---|---|
| $512 \times 512$ resolution | | | | | |
| Adam | 0.831 | 0.837 | 0.806 | 0.484 | 0.834 |
| AdamW | 0.337 | 0.870 | 0.349 | 0.245 | 0.485 |
| SGD | 0.964 | 0.949 | 0.976 | 0.801 | 0.956 |
| $640 \times 640$ resolution | | | | | |
| Adam | 0.818 | 0.762 | 0.785 | 0.430 | 0.789 |
| AdamW | 0.927 | 0.807 | 0.875 | 0.678 | 0.863 |
| **SGD** | **0.966** | **0.957** | **0.979** | **0.830** | **0.961** |
| $1024 \times 1024$ resolution | | | | | |
| Adam | 0.818 | 0.762 | 0.785 | 0.430 | 0.789 |
| AdamW | 0.919 | 0.720 | 0.806 | 0.620 | 0.808 |
| SGD | 0.964 | 0.968 | 0.973 | 0.826 | 0.966 |

Looking at mAP@0.95, it is observed that the SGD optimizer with a resolution of $640 \times 640$ achieved the best results among all the models, reaching a value of 0.83. With mAP@0.95, mAP@0.5 for the same model is 0.979. On the other hand, the YOLOv5 model with AdamW optimizer achieved the lowest results, with mAP@0.95 of 0.34 and mAP@0.5 of 0.245. When it comes to precision and recall criteria, the SGD optimizer stands out again, achieving the best results compared to other models obtained at a resolution of $640 \times 640$. In this case, the achieved precision of 0.966 represents the highest result achieved with regard to this metric. Regarding the recall, the SGD optimizer also achieves the best results, but at a different resolution, namely $1024 \times 1024$. The F1 score reaches its maximum value in the case of the SGD optimizer at a resolution of $1024 \times 1024$, which is equivalent to 0.966.

On the other hand, the AdamW optimizer achieves the lowest F1 score at a resolution of $512 \times 512$, which is 0.485. Of course, in addition to the mentioned metrics, it is important to carefully study the confusion matrix, also known as the responsibility matrix, which was carried out. In this context, the best model for each individual resolution or the best optimizer for each image resolution was carefully filtered. The selection of the best model is based on the evaluation measure mAP@0.95.

According to the representation in Figure 13, it can be observed that the individual value of each element of the responsibility matrix exceeds the 0.9 threshold, which emphasizes the high degree of responsibility of the model predictions. In particular, the smallest coefficient of responsibility can be noted for the TA class, with an amount of the coefficient of responsibility of 0.91. This indicates the precision of detection for all classes, and especially for the problematic TA class, where the model showed an exceptional degree of accuracy. In the context of incorrect class prediction, we note that all coefficients are located on the main diagonal of the liability matrix. This feature indicates that the model did not produce an incorrect classification during the evaluation, as all detected objects were correctly classified. The absence of values outside the main diagonal further confirms the

absence of any irregularities or incorrect class predictions. These insights emphasize the exceptional precision of the model in object detection and confirm the reliability of the algorithm, especially in the identification of the TA class.

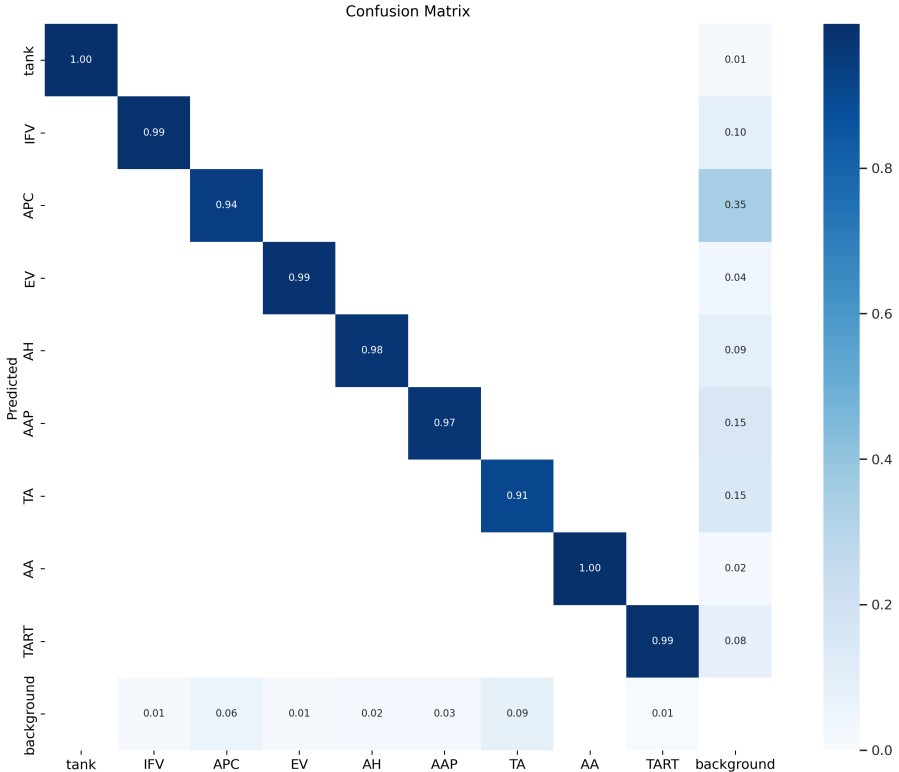

**Figure 13.** The responsibility matrix for SGD optimizer and 512 × 512 resolution.

In contrast to the previously analyzed case, upon careful observation of Figure 14, subtle but relevant differences are visible in the results. These differences are manifested in the dynamics of object detection performance between the two experimental contexts. Specifically, there are variations in the responsibility coefficients for different classes. Some of the classes experienced an increase in performance, which is evident in the case of the EV class, while at the same time a decrease in performance was recorded for the TA and AAP classes, which is manifested by a decrease in the coefficients of responsibility for these classes.

Furthermore, a deeper analysis of these variations indicates the importance of a detailed investigation of the causes of these changes. Possible factors that contributed to the growth of responsibility coefficients for certain classes include model optimization, improved data representation, or fine-tuning of hyperparameters. On the other hand, a drop in performance for certain classes may arise from challenges in detecting those specific objects, possibly due to variability in their characteristics or scene context.

These findings highlight the importance of continuous monitoring and evaluation of model performance, which includes analyzing detected objects against real objects. Understanding these nuances is crucial for the iterative development and improvement of object detection algorithms, and opens up opportunities for further optimization in order to achieve reliable and precise detection in different application contexts.

Finally, the SGD optimizer is evaluated in Figure 15 with a resolution of 1024 × 1024. The results of mAP@0.95 showed that the coefficients of responsibility had significant values for all detected objects, implying that the model predictions were highly reliable in this context. Similarly to the two previous cases, no irregularities in class predictions were observed. This characteristic of the model further confirms its capability to accurately detect objects at high resolution. The absence of incorrect predictions or misclassified

objects demonstrates the high accuracy and dependability of the model in this particular resolution setting.

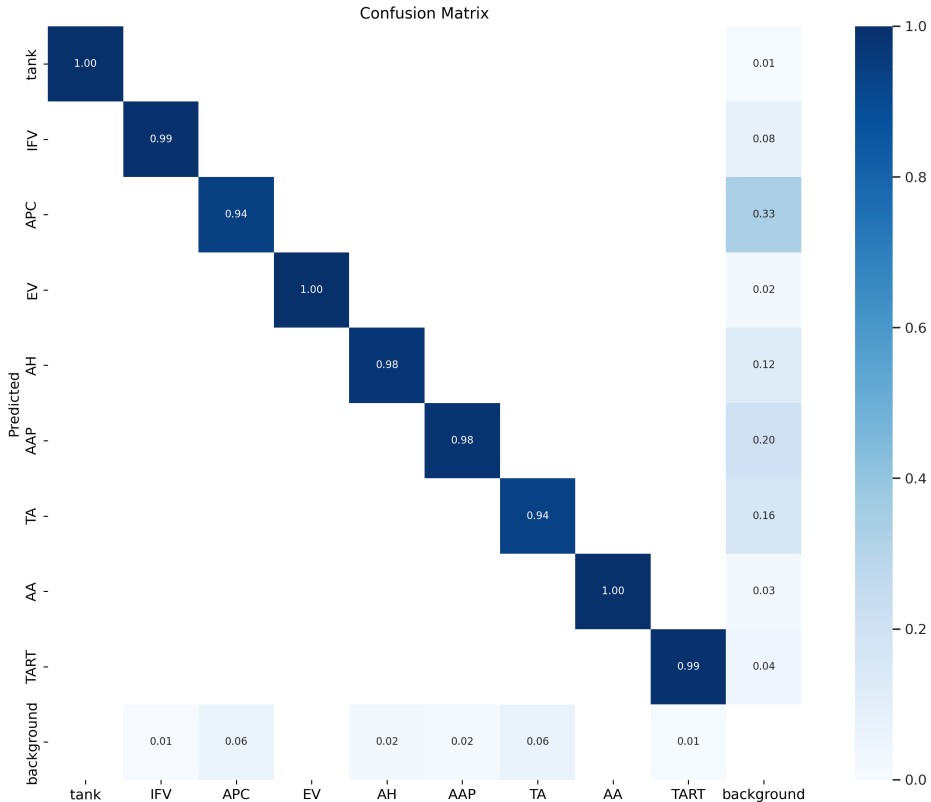

**Figure 14.** The responsibility matrix for SGD optimizer and $640 \times 640$ resolution.

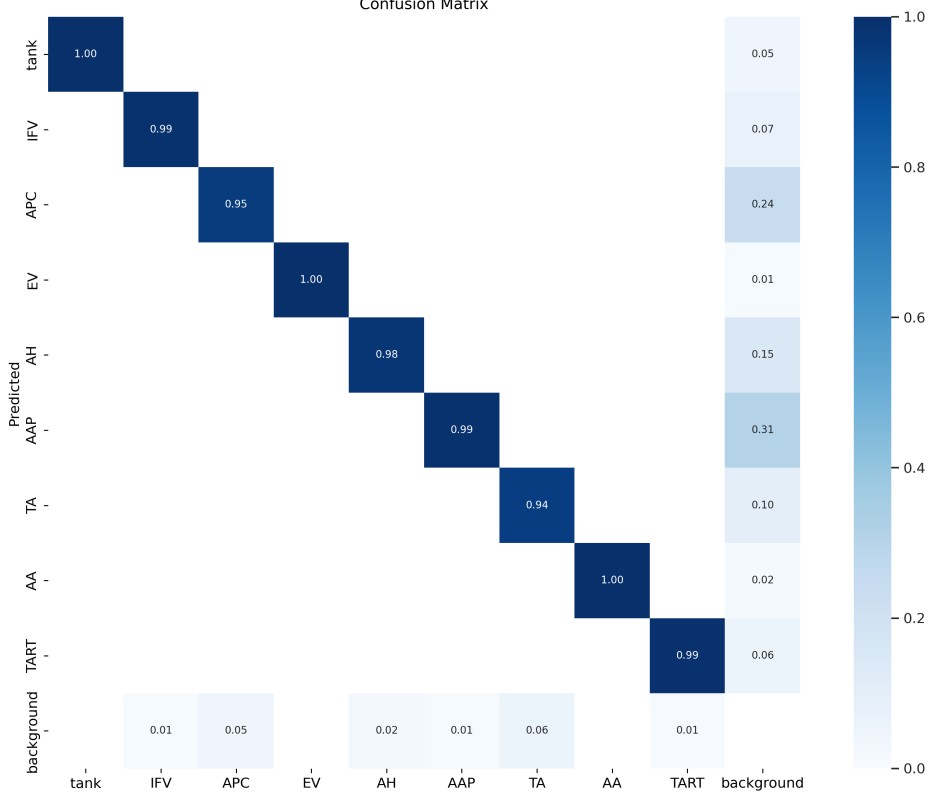

**Figure 15.** The responsibility matrix for SGD optimizer and $1024 \times 1024$ resolution.

After a thorough examination of the confusion matrix, which provides insight into the accuracy of the detection of enemy objects, the focus shifts to the key factor of the system's efficiency: the speed of interference. This speed, which is the rate at which the algorithm identifies objects in scenes, is of great importance in military operations, particularly when it comes to UAVs. In such dynamic settings, the speed of enemy detection can be a deciding factor in making real-time operational decisions. YOLOv5 is an algorithm that is both precise and fast. The speed of the algorithm was tested on a laptop and a desktop computer, the specifications of which are listed in Table 3.

**Table 3.** Technical specifications of the laptop computer.

| Operating System | Windows 11 Pro |
|---|---|
| CPU | 11th Gen Intel Core i7–1195G7 @2.9 GHz |
| GPU | Intel Iris Xe Graphics |
| RAM | 32 GB 3200 MHz DDR4 |

The specifications of the desktop computer on which the model was trained are shown in Table 4.

**Table 4.** Technical specifications of the desktop computer.

| Operating System | Windows 11 Pro |
|---|---|
| CPU | Ryzen 7 5800X 8 Cores up to 4.7 GHZ |
| GPU | Nvidia GeForce RTX 3070 8 GB |
| RAM | 32 GB 3000 MH DDR4 |

The results of the detection time are presented in Table 5. An integrated GPU yielded an interference value of 168.5 ms, while a conventional GPU produced a value of 24 ms. These findings are shown in the table.

**Table 5.** Average interference time for $640 \times 640$ resolution using best model from this study.

| System | Average Interference Time [ms] |
|---|---|
| Laptop computer | 168.5 |
| Desktop computer | 24 |

While conducting military operations, the ability to quickly detect the enemy is of the utmost importance. YOLOv5 offers a rapid response to any changes that may occur in the field. This could significantly enhance operational capabilities, allowing better understanding of the environment and making it possible to make informed decisions in real time. The significance of employing the YOLOv5 algorithm for military object detection is underscored by the insights provided in Table 6. The weight of the obtained YOLOv5 model, ranging from 13 to 14 megabytes, is notably compact, presenting opportunities for deployment on resource-constrained devices, such as Raspberry Pi [47–49] . However, it is crucial to note that for efficient data processing and inference, a moderately powerful graphics unit, particularly one containing a GPU, is still mandatory.

**Table 6.** YOLOv5 weights of obtained models.

| Optimizer | Resolution [Height × Width] | Size [MB] |
| --- | --- | --- |
| Adam | 512 × 512 | 13.6 |
| Adam | 640 × 640 | 13.7 |
| Adam | 1024 × 1024 | 13.8 |
| AdamW | 512 × 512 | 13.6 |
| AdamW | 640 × 640 | 13.7 |
| AdamW | 1024 × 1024 | 13.8 |
| SGD | 512 × 512 | 13.6 |
| SGD | 640 × 640 | 13.7 |
| SGD | 1024 × 1024 | 13.8 |

## 4. Discussion

An examination of the mAP@0.95 results revealed an outstanding performance of the SGD optimizer at a resolution of 640 × 640, with a remarkable mAP@0.95 score of 0.83. In comparison, the YOLOv5 model with the AdamW optimizer had the poorest performance, attaining only 0.34 for mAP@0.95 and 0.245 for mAP@0.5. The SGD optimizer was particularly impressive with its high precision (0.966) and responsiveness values, with the best result achieved at a resolution of 1024 × 1024. The F1 score also registered the highest value of the SGD optimizer at a resolution of 1024 × 1024 (0.966). On the other hand, the AdamW optimizer achieved the lowest F1 score at a resolution of 512 × 512 (0.485). Analysis of the confusion matrix (responsibility matrix) further confirmed the high accuracy in object detection, particularly for the critical TA class. There were no misclassifications, which further attests to the high precision and dependability of the model. The detection results were optimal in both cases, whether using a laptop or a desktop computer. Although the laptop does not have an advanced graphics system, it still achieved surprisingly good results. However, using a desktop computer, there was a considerable decrease in detection time, often even several times, which is clearly visible in Table 5.

The results obtained and a more thorough analysis demonstrate the need for ongoing monitoring and assessment of model performance, as well as further research into the reasons for the variations in the results. Comprehending these nuances is essential for improving object detection algorithms and achieving high detection accuracy in different application contexts. Despite the average results obtained through mAP, recall, precision, and other metrics, a more comprehensive understanding of the evaluation state is obtained from the analysis of confusion matrices. These matrices provide detailed information on the distribution of predictions between classes and show the connection between actual and predicted values. For example, although the average mAP@0.95 results indicated different optimizer performance and resolutions, the confusion matrices revealed certain classes that may have had an effect on these results. It is possible that the detection precision of a critical class, such as TA, had a major influence on the average metrics. Specifically, high accuracy in the detection of these key classes can lead to better overall results.

The time frame of interference is critical for the operability of a system used in military operations and reconnaissance. Generally, disturbances in electronic or signal communications should remain within a few tens of milliseconds. This is essential for the success of scouting activities and for the safety of the personnel involved. Quick decisions must be made during reconnaissance missions, and any delay or prolonged interruption in communication or data collection could be detrimental. The data collected must be quickly and accurately analyzed to provide timely information for strategic planning and implementation. Meeting the technical parameters is necessary for efficient processing and utilization of the information, but the final evaluation of the efficiency and effectiveness of the system

is performed by experienced personnel in the defense forces. These professionals have the expertise and knowledge of the context in which the system is used, making their assessment essential for the system's functionality and suitability for military operations.

## 5. Conclusions

This research presents a detailed analysis of the training of different YOLOv5 models using different optimizers and image resolutions. Research results emphasize that it is not always possible to achieve optimal performance by increasing the image resolution during training. Different image resolutions and optimizers have shown different effects on model performance, and standard evaluation metrics do not always provide the most accurate insight into actual model performance. Evaluation metrics such as mAP, recall, and precision are useful, but not always the most faithful representation of the model, especially in the context of object detection in military applications. Therefore, it is advisable to consider more aspects and methods of evaluation in order to get a more complete picture of the performance of the model. During the testing, limited detection of objects was observed, which is not satisfactory for military purposes. However, the analysis of the response values (recall) indicates the possibility of further research into the application of computer vision on aircrafts to detect objects on the ground.

In conclusion, it is of utmost importance to address the hypothetical questions posed at the outset of this work, namely:

- That it is possible through thorough research and study of multiple image materials to develop a sufficiently high-quality dataset that will be used to train an artificial intelligence model, or in this case, a detection algorithm;
- That it is possible to detect, classify, and localize objects such as flying objects, mobile objects, etc., of military purpose by applying well-developed models such as YOLOv5;
- The completed methodology manifests qualitative results, with the application of which commanders of the armed forces can make decisions of considerable responsibility, eliminating occurrences of undesirable consequences. At the same time, taking into account resources of lower performance on equipment that does not require high performance, decision-making is approached with optimal efficiency.

For future research, it is suggested to analyze the effect of image resolution on the model's evaluation values. Additionally, it is essential to balance the entire dataset to create better quality models that would be more suitable for military purposes. Implementing the model on a spacecraft and testing its performance in real-world conditions is of great importance for furthering research in this field. Furthermore, researching new techniques and algorithms that would enable more accurate detection of objects on the ground could be critical to improving military applications of computer vision. Moreover, including humans (soldiers or civilians) into one or more classes would be highly beneficial because of their importance in modern combat. Finally, it is of utmost importance to implement the model and assess its performance in real-world scenarios. The evaluation can be conducted using videos obtained by commercial UAVs equipped with high-resolution payloads, such as 4K UHD 2160p. However, optimal evaluation would involve the use of a military-grade payload due to its relevance in real-world applications.

**Author Contributions:** Conceptualization; N.Ž., M.G. and I.L.; methodology; N.Ž. and D.M.; software; M.G. and I.L.; validation; N.Ž., M.G., D.M. and I.L.; formal analysis; N.Ž. and M.G.; investigation; I.L. and D.M.; resources; N.Ž. and M.G.; data curation; D.M. and I.L.; writing—original draft preparation; N.Ž. and M.G.; writing—review and editing; N.Ž., M.G. and I.L.; visualization; D.M. and I.L.; supervision; I.L. and D.M.; project administration; M.G., I.L. and D.M.; funding acquisition; N.Ž. and M.G. All authors have read and agreed to the published version of the manuscript.

**Funding:** This research received no external funding.

**Data Availability Statement:** Not applicable.

**Conflicts of Interest:** The authors declare no conflict of interest.

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
