# Peer review of "Military Decision-Making Process Enhanced by Image Detection"

_information, doi:10.3390/info15010011_

Round 1

Reviewer 1 Report

Comments and Suggestions for Authors

The manuscript investigates the use of YOLOv5 detection framework for military equipment detection. A dataset was curated for this purpose by collecting public data. Details of data collection, model training, hyper-parameter tuning are presented and investigated towards optimal detection accuracy in this specific application context. Via extensive experiments, a set of recommended optimal hyper-parameters and training strategies are given and limitations are discussed for potential future work. As a research paper, there exist issues to be addressed before publication.

-- The abstract needs to be rewritten. It should contain the description of research background, motivations, methods, results and conclusions in brevity. The current abstract even fails to convey the resesarch problem itself.

-- Similar to the issues of abstract, the introduction section also fails to give the most relevant information to the research, i.e., military equipment detection using YOLOv5. Instead, most content in the current version of introduction are redundant and hence should be removed.

-- In section 2.1, the description of each military equipment is redundant (i.e. lines118-150).

-- In Figure 2, bounding boxes should be given in the exemplar images.

-- Description in lines 185-212 gives the motivation of research and problem formulation and hence should be presented in an earlier section (e.g., introduction).

-- What's the reason of giving information in table 1? Is it formulated as a 3-class object detection problem?

-- Please justify why YOLOv5 is chosen in the study. Why not use more advanced detection frameworks, e.g., DETR? How about other versions of YOLO and other classical detectors such as Faster RCNN, SSD, RetinaNet and etc.?

-- In summary, the study is incomprehensive and more works are needed for a thorough comparative study.

Comments on the Quality of English Language

The organization of this paper needs to be improved significantly.

Author Response

Respected Reviewer,

We would like to express our sincere gratitude for your valuable comments and insights. Your feedback has been instrumental in enhancing the quality and clarity of our manuscript. We have carefully considered all your suggestions and marked the implemented changes in red throughout the manuscript. Your constructive input has significantly contributed to the improvement of our work.

Thank you once again for your time and thoughtful review.

Best regards

The Authors

Comments of the reviewer

The manuscript investigates the use of YOLOv5 detection framework for military equipment detection. A dataset was curated for this purpose by collecting public data. Details of data collection, model training, hyper-parameter tuning are presented and investigated towards optimal detection accuracy in this specific application context. Via extensive experiments, a set of recommended optimal hyper-parameters and training strategies are given and limitations are discussed for potential future work. As a research paper, there exist issues to be addressed before publication.

Comment 1 

 The abstract needs to be rewritten. It should contain the description of research background, motivations, methods, results and conclusions in brevity. The current abstract even fails to convey the resesarch problem itself.

Answer

According to reviewers comment abstract is rewritten into:

This study delves into the vital missions of the Armed Forces, encompassing the defense of territorial integrity, sovereignty, and support for civil institutions. Commanders grapple with crucial decisions, where accountability underscores the imperative for reliable field intelligence. Harnessing artificial intelligence, specifically the YOLO version five detection algorithm, ensures a paradigm of efficiency and precision. The presentation of trained models, accompanied by pertinent hyperparameters and dataset specifics derived from public military insignia videos and photos, reveals a nuanced evaluation. Results, scrutinized through precision, recall, map@0.5, mAP@0.95 and F1 score metrics, illuminate the supremacy of the model employing Stochastic Gradient Descent at 640 x 640 resolution: 0.966, 0.957, 0.979, 0.830, and 0.961. Conversely, the suboptimal performance of the model using the Adam optimizer registers metrics of 0.818, 0.762, 0.785, 0.430, and 0.789. These outcomes underscore the model's potential for military object detection across diverse terrains, with future prospects considering implementation on unmanned aerial vehicles to amplify and deploy the model effectively.

Comment 2 

Similar to the issues of abstract, the introduction section also fails to give the most relevant information to the research, i.e., military equipment detection using YOLOv5. Instead, most content in the current version of introduction are redundant and hence should be removed.

Answer

The authors agree with reviewers comments and we attach the manuscript modifications:

rows 38 to 43 are removed,

rows 58-68 were removed,

rows78-92 has been reconstructed and now reads:

IMINT, an abbreviation for Imagery Intelligence, is pivotal in contemporary data collection, harnessing information from electro-optical, infrared, RADAR, and LIDAR sensors across a variety of platforms such as land, sea, air, and space platforms. It is important to highlight that humans using any type of camera can also gather IMINT data [11]. Unmanned Aerial Vehicles (UAV) stand out for their significant potential in diverse missions within modern warfare and homeland security systems [17]. Post-data collection, the information is meticulously processed and presented as refined images or videos for end users. The importance of IMINT is underscored by its detailed categorization, as articulated in the NATO doctrine, recognizing 19 distinct categories [18]. Analysts, each specialized in a specific category, process gathered data and create intelligence products. However, challenges arise when gathered data includes multiple categories, demanding considerable interpretation time. In scenarios requiring swift processing, the integration of Artificial Intelligence (AI), exemplified by You Only Look Once (YOLO), proves invaluable. This approach allows AI to efficiently process comprehensive data, with analysts subsequently verifying the results. The significance of these data collection methods lies in their capability to provide detailed insights promptly, contributing to timely decision-making in dynamic operational environments.

The review of the literature or State of The Art has now been modified and in the new version of the manuscript it reads:

”Building upon the information at hand, and discerning the data source is achievable through the diverse resources mentioned earlier. Nonetheless, a pivotal inquiry arises concerning the identification of a spectrum of military installations. Bearing this in mind, numerous scholars have undertaken diverse research endeavors in this domain to delve more deeply into various methodologies for detecting adversary objects. Pham and Polasek  [19] investigated the development of a lightweight military target detection method, SMCA- YOLOv5. The method, which involves replacing the focusing module and redesigning the network structure, achieves an exceptional result with an average accuracy of 98.4% and a detection speed of 47.6 FPS. It outperforms competing algorithms such as SSD, and Faster- RCNN, with a significant reduction in parameter cardinality and computational burden compared. In their research, Hu et al. [20] used the method of Optimal Gabor Filtering and Deep Feature Pyramid Network to utilize a military target detection dataset named MOD VOC, which was created to meet the PASCAL VOC dataset format standard and includes images primarily sourced from video footage captured by unmanned aerial vehicles (UAVs), ground cameras, and internet images. In doing so, they used five artificial intelligence algorithms (Faster R-CNN, DSOD300, DSSD513, YOLOv2 544, and their filtering method) to compare the individual performance of the best model for each algorithm. The best results for MOD VOC in terms of accuracy, recall, and average fps were achieved by their model in the amount of 88.76%, 78.45%, and 30.35 respectively, while the worst results were achieved by DSSD513 in the amount of 73.57%, 64.56% and 42.43. Kong et al. [21] used a military target dataset showing armed individuals with different weapons to improve the detection performance of the proposed YOLO-G algorithm. The authors introduced improvements compared to the YOLOv3 framework, including a lightweight GhostNet for improved accuracy and speed in detecting military targets. The dataset evaluation showed a 2.9% improvement in mAP and a 25.9 FPS increase in detection rate compared to the original YOLOv3, highlighting the effectiveness of their improved algorithm. Wang and Han [22] introduce the YOLO-M algorithm for military equipment target recognition, addressing challenges in small target detection. By incorporating the C3CMix module and modifying the activation function in YOLOv5, the proposed algorithm maintains high accuracy while reducing parameters, resulting in a 95.2% average accuracy, an 18.8% reduction in parameters, and a 14.5% decrease in computation. These improvements make YOLO-M well-suited for deployment in military equipment target recognition applications. Du et al. [23] investigated military vehicle object detection based on hierarchical feature representation and refined localization for the detection of military objects in the desert, grass, snow, city, and others. The authors applied R-FCNN, SSD, YOLOv3, YOLOv4, Faster R-CNN, and MVODM, i.e. a novelty algorithm created by the author. The models were trained on three different types of test data sets, i.e. large-scale, small-scale, and all subset test data sets. the best results were shown by MVODM (novelty algorithm) for a large-scale data set with evaluation metrics results in the amount of 85.6% mAP, while the worst performance was YOLOv3 for a small subset of data in the amount of 54.9% mAP. Nelson and McDonald [24] developed the Multisensor Towed Array System (MTADS), demonstrating its effectiveness in detecting buried unexploded ordnance with an outstanding probability of detection (0.95 or better). The system results highlight its precision in locating ordnance at self-penetrating depths, providing a cost-effective and accelerated approach to remediation compared to standard techniques. Pham and Polasek [19] address the challenges of surface object detection in urban environments, utilizing both infrared and visible spectra. The paper aims to develop an algorithm for detecting and selecting objects of interest, particularly civil automobiles resembling military equipment, captured by infrared and visible cameras in various outdoor conditions. The proposed algorithm involves determining optimal threshold values for image conversion in changing environmental conditions and colors, tested on static images, and extended to dynamic object detection, selection, and tracking through video processing. Additionally, the study emphasizes the use of threshold adjustment techniques to optimize object detection. Qin et al. [25] introduce a method inspired by EfficientDet trackers for classifying maritime military targets in high-resolution optical remote sensing images. The approach involves constructing a multilayer feature extraction network with attention mechanisms, utilizing ReLU activation, and employing deep feature fusion networks and prediction networks to accurately identify various types of military ships. The trained model was tested on six classes of vessels, achieving the best detection performance in the GW class with precision and recall values of 0.983 and 0.945, respectively. The lowest detection results were observed in the SS class, with precision and recall values of 0.974 and 0.822, respectively.”

Comment 3 

In section 2.1, the description of each military equipment is redundant (i.e. lines 118-150).

Answer

According to the comment, lines 118-150 are removed, and only the list of classes remains in the text. But we still think the description of each class is useful for readers to understand each class and it’s role in the modern battlefield.

Comment 4 

In Figure 2, bounding boxes should be given in the exemplar images.

Answer

Figure 2. from the previous version of the manuscript has been changed with a new one, on which the ground truth locations and detections of objects of a military nature are labeled.

Comment 5 

Description in lines 185-212 gives the motivation of research and problem formulation and hence should be presented in an earlier section (e.g., introduction).

Answer

According to this comment, this section is moved to Introduction.

Comment 6 

What's the reason of giving information in table 1? Is it formulated as a 3-class object detection problem?

Answer

The reason for giving information in table 1 is to highlight the fact there is a NATO UAS classification which classifies all UAS into three different classes. Main difference between classes is UAS’ MTOW (maximum takeoff weight). However classes also distinguish in terms of operational altitude, operational range, payload quality and mass and so on. It is also important to highlight that UAS operators for classes II and III usually fly UAV from the comfort of their base or another type of military outpost, while class I operators actually go in the field and spend hours while executing missions. Moreover, class I operators are exposed to more stress and accumulate a lot of fatigue during mission execution (missions can last up to 6 or more hours). Therefore the table’s purpose is to highlight mentioned facts to the reader who is not an expert in UAS classification. In order to present this as clear as possible, table is deleted and it’s content is described with following sentences:

“Considering intelligence data collection from IMINT sources, special attention in this paper will be given to collection with unmanned aerial systems (UAS) and challenges that emerge while collecting and analyzing data collected with UAS. Current NATO UAS classification recognizes three different UAS classes. [25] Those are classes I, II and III. Main class features these classes are differentiated by is their maximum takeoff weight (MTOW). Class I includes UAS whose MTOW doesn’t exceed 150 kg. Class II includes UAS whose MTOW is more than 150 kg, but less than 600 kg. Lastly, class III includes UAS whose MTOW exceeds 600 kg. Bigger MTOW value means UAS is capable of carrying a larger amount of payload and therefore be able to execute missions which are more complex. Moreover, mentioned classes also differentiate by level of command they are subordinated to, their operating altitude, their range and their operating range. It is important to highlight that the flying crew of classes II and III consist of two crew members. First one is a pilot and is responsible for execution of flying operations with unmanned aerial vehicles (UAV) while the second member is a payload operator, responsible for gathering IMINT data. On the other hand UAS class I systems are operated by only one operator, who is a pilot and also a payload operator. This means that class I operators are exposed to rather high mental workload, in terms of multitasking (operating sensor and watching all flight parameters at the same time) during the whole mission which may last up to 6 hours. This high mental workload results in a lot of accumulated fatigue which makes UAS operator less concentrated in later stages of mission. In order to make UAS class I operators’ work easier, using algorithms with small detection time, such as YOLOv5 would be highly beneficial. It would detect objects of interest instead of operator and he would only confirm with a bigger zoom if that is really the object of interest or he could move on. Moreover, the operator could concentrate more on flying parameters and noticing any malfunction. Furthermore, another issue that may emerge is that units headquarters and all subordinate units don't have enough capacity to analyze all gathered IMINT data or just don't have enough time to do it. Current practice for IMINT data analysis is data being analyzed by an IMINT analyst who is narrowly specialized in one of 19 different IMINT categories according to NATO doctrine. Sometimes IMINT data may consist of different elements which belong to different categories. This may pose a problem during complex operations which require fast flow of information, especially if a country's Armed Forces do not have an IMINT analyst specialized in each class and one analyst has to analyze a few different classes. YOLOv5 is an ideal solution for this problem due to its speed, reliability, and accuracy. It is capable of handling various challenges in the detection, classification and segmentation of objects”

Comment 7 

Please justify why YOLOv5 is chosen in the study. Why not use more advanced detection frameworks, e.g., DETR? How about other versions of YOLO and other classical detectors such as Faster RCNN, SSD, RetinaNet and etc.?

Answer

The selection of YOLOv5 in the paper can be justified by several facts. YOLOv5 is known for its efficiency, speed, and accuracy, which makes it suitable for real-time object detection tasks. Its architecture has seen significant improvements over earlier versions, addressing limitations and improving performance. The decision to use YOLOv5 instead of more advanced frameworks such as DETR or other classical detectors such as Faster RCNN, SSD and RetinaNet can be significantly more complex, and further below are additional justifications that we consider YOLOv5 to be the optimal choice for this research work: 

Efficiency and Speed- YOLOv5 is renowned for its fast and efficient object detection capabilities, making it well-suited for scenarios where real-time processing is crucial. The study may prioritize speed and efficiency over some of the more complex but computationally demanding models like DETR.

Architectural improvement - YOLOv5 represents an evolution in the YOLO architecture, containing improvements and optimizations. These enhancements make it a competitive choice for certain applications, especially when considering a balance between accuracy and speed.

Task-specific considerations- The specific characteristics of the object detection task in the study align well with the strengths of YOLOv5. For instance, the dataset consists of numerous small objects and real-time performance is critical.

Simplicity of possibility for implementation- YOLOv5 is praised for its simplicity and ease of implementation, making it an attractive choice for researchers and practitioners who want a straightforward yet effective solution. Also, it opens a possibility for implementation on trivial microcontrollers such as Raspberry Pi.

Also in the methodology section, following text is added:

”The YOLOv5 detection algorithm excels in achieving an optimal balance between speed and accuracy, featuring a refined architecture suitable for implementation on resource-constrained microcontrollers [30]. From the compact YOLOv5 nano model to larger variants, these models exhibit memory weights conducive to diverse applications, including military contexts. Comparative evaluations against DETR and EfficientDet underscore YOLOv5’s superiority, particularly in challenges such as crop circle detection, where it outperforms with a recall of 0.98, surpassing DETR and EfficientDet 0.68 and 0.85, respectively, alongside precision values of 0.77 and 0.91 [30–32]. In scenarios involving overlapping object detection in kitchens, YOLOv5 excels by producing accurate frames and demonstrates superior performance compared to Faster R-CNN [33]. The study reveals YOLOv5’s accuracy of 0.8912 (89.12%) outshining Faster R-CNN’s 0.8392 (83.92%), underscoring its effectiveness in handling complex scenarios. Beyond object detection, YOLOv5 exhibits computational efficiency, outpacing sophisticated methods like RetinaNet, as observed [34–36]. Despite not universally achieving optimal results, the model’s performance nuances are vital considerations in addressing computational complexities during training.The versatility of YOLOv5 extends across domains, including the analysis of secondary waste treatment processes, applications in autonomous vehicles, and weed growth detection. Particularly noteworthy is its exceptional interference reduction and efficiency in weed growth detection, positioning YOLOv5 as an optimal choice for military applications, as emphasized [37]. Owing to foundational principles and analogous use cases, YOLOv5, characterized by its intricate, comprehensive, precise, and rapid attributes, undergoes scrutiny for the detection of military objects.”

Comment 8 

In summary, the study is incomprehensive and more works are needed for a thorough comparative study.

Answer

The authors concur with the aforementioned assertion, elaborated upon in the introduction of this study. Furthermore, the literature review is expounded upon in the methodology section, incorporating a comparison and underscoring the significance of employing the YOLOv5 algorithm for intricate tasks.

A modified literature review that can be compare with this work is now added in the new version of the manuscript and now is as follows:”

"Building upon the information at hand, and discerning the data source is achievable through the diverse resources mentioned earlier. Nonetheless, a pivotal inquiry arises concerning the identification of a spectrum of military installations. Bearing this in mind, numerous scholars have undertaken diverse research endeavors in this domain to delve more deeply into various methodologies for detecting adversary objects [19] investigated the development of a lightweight military target detection method, SMCA--YOLOv5. The method, which involves replacing the focusing module and redesigning the network structure, achieves an exceptional result with an average accuracy of 98.4% and a detection speed of 47.6 FPS. It outperforms competing algorithms such as SSD, and Faster-RCNN, with a significant reduction in parameter cardinality and computational burden compared. In their research, [20] used the method of Optimal Gabor Filtering and Deep Feature Pyramid Network to utilize a military target detection dataset named MOD VOC, which was created to meet the PASCAL VOC dataset format standard and includes images primarily sourced from video footage captured by unmanned aerial vehicles (UAVs), ground cameras, and internet images. In doing so, they used five artificial intelligence algorithms (Faster R-CNN, DSOD300, DSSD513, YOLOv2 544, and their filtering method) to compare the individual performance of the best model for each algorithm. The best results for MOD VOC in terms of accuracy, recall, and average fps were achieved by their model in the amount of 88.76%, 78.45%, and 30.35 respectively, while the worst results were achieved by DSSD513 in the amount of 73.57%, 64.56%, and 42.43. Kong et al. [21] used a military target dataset showing armed individuals with different weapons to improve the detection performance of the proposed YOLO-G algorithm. The authors introduced improvements compared to the YOLOv3 framework, including a lightweight GhostNet for improved accuracy and speed in detecting military targets. The dataset evaluation showed a 2.9% improvement in mAP and a 25.9 FPS increase in detection rate compared to the original YOLOv3, highlighting the effectiveness of their improved algorithm. Wang and Han [22] introduces the YOLO-M algorithm for military equipment target recognition, addressing challenges in small target detection. By incorporating the C3CMix module and modifying the activation function in YOLOv5, the proposed algorithm maintains high accuracy while reducing parameters, resulting in a 95.2% average accuracy, an 18.8% reduction in parameters, and a 14.5% decrease in computation. These improvements make YOLO-M well-suited for deployment in military equipment target recognition applications. Du et al. [23] investigated military vehicle object detection based on hierarchical feature representation and refined localization for the detection of military objects in the desert, grass, snow, city, and others. The authors applied R-FCNN, SSD, YOLOv3, YOLOv4, Faster R-CNN, and MVODM, i.e. a novelty algorithm created by the author. The models were trained on three different types of test datasets, i.e. large-scale, small-scale, and all subset test datasets. The best results were shown by MVODM (novelty algorithm) for a large-scale dataset with evaluation metrics results in the amount of 85.6% mAP, while the worst performance was YOLOv3 for a small subset of data in the amount of 54.9% mAP. Nelson and McDonald [24] developed the Multisensor Towed Array System (MTADS), demonstrating its effectiveness in detecting buried unexploded ordnance with an outstanding probability of detection (0.95 or better). The system results highlight its precision in locating ordnance at self-penetrating depths, providing a cost-effective and accelerated approach to remediation compared to standard techniques. Pham and Polasek [19] addresses the challenges of surface object detection in urban environments, utilizing both the infrared and visible spectra. The paper aims to develop an algorithm for detecting and selecting objects of interest, particularly civil automobiles resembling military equipment, captured by infrared and visible cameras in various outdoor conditions. The proposed algorithm involves determining optimal threshold values for image conversion in changing environmental conditions and colors, tested on static images, and extended to dynamic object detection, selection, and tracking through video processing. Additionally, the study emphasizes the use of threshold adjustment techniques to optimize object detection [25]. introduces a method inspired by EfficientDet trackers for classifying maritime military targets in high-resolution optical remote sensing images. The approach involves constructing a multilayer feature extraction network with attention mechanisms, utilizing ReLU activation, and employing deep feature fusion networks and prediction networks to accurately identify various types of military ships. The trained model was tested on six classes of vessels, achieving the best detection performance in the GW class with precision and recall values of 0.983 and 0.945, respectively. The lowest detection results were observed in the SS class, with precision and recall values of 0.974 and 0.822, respectively."

Reviewer 2 Report

Comments and Suggestions for Authors

Military decision-making process enhanced by image detection is proposed and it looks good. However, it requires some modifications.

1. please include quantitative results in the abstract for indicating superiority of your approach.

2. Please introduce more image detection techniques.

3. It will be better to introduce some research questions addressed by your research.

4. method is not novel. Please justify its importance for the dataset used.

5. Please give detailed analysis about the methods and parameter tuning along with proper descriptions about the dataset.

6. it will be better to compare the results of your approach with more recent methods in the literature.

7. it will be better to highlight the best results in the experimental results to indicate superiority of proposed approach.

8. It will be better to edit the manuscript with the native speaker to improve the quality of English.

Comments on the Quality of English Language

moderate English edit is required.

Author Response

Respected Reviewer,

We want to extend our sincere appreciation for your invaluable comments and insights. Your feedback has played a crucial role in improving the quality and clarity of our manuscript. We have taken careful note of all your suggestions and indicated the implemented changes in cyan throughout the manuscript. Your constructive input has greatly contributed to refining our work.

We thank you once again for dedicating your time to providing thoughtful feedback.

Best regards

The Authors

Comments of the Reviewer

Military decision-making process enhanced by image detection is proposed and it looks good. However, it requires some modifications.

Comment

  1. please include quantitative results in the abstract for indicating superiority of your approach. 

Answer 

In the new version of the manuscript the abstract was modified, and now is written as: 

This study delves into the vital missions of the Armed Forces, encompassing the defense of territorial integrity, sovereignty, and support for civil institutions. Commanders grapple with crucial decisions, where accountability underscores the imperative for reliable field intelligence. Harnessing artificial intelligence, specifically the YOLO version five detection algorithm, ensures a paradigm of efficiency and precision. The presentation of trained models, accompanied by pertinent hyperparameters and dataset specifics derived from public military insignia videos and photos, reveals a nuanced evaluation. Results, scrutinized through precision, recall, map@0.5, mAP@0.95 and F1 score metrics, illuminate the supremacy of the model employing Stochastic Gradient Descent at 640 x 640 resolution: 0.966, 0.957, 0.979, 0.830, and 0.961. Conversely, the suboptimal performance of the model using the Adam optimizer registers metrics of 0.818, 0.762, 0.785, 0.430, and 0.789. These outcomes underscore the model's potential for military object detection across diverse terrains, with future prospects considering implementation on unmanned aerial vehicles to amplify and deploy the model effectively.

Comment 2

  1. Please introduce more image detection techniques. 

Answer

In the introduction of this manuscript, a more detailed overview of the literature is provided, which reads:

Building upon the information at hand, and discerning the data source is achievable through the diverse resources mentioned earlier. Nonetheless, a pivotal inquiry arises concerning the identification of a spectrum of military installations. Bearing this in mind, numerous scholars have undertaken diverse research endeavors in this domain to delve more deeply into various methodologies for detecting adversary objects. Pham and Polasek  [19] investigated the development of a lightweight military target detection method, SMCA-YOLOv5. The method, which involves replacing the focusing module and redesigning the network structure, achieves an exceptional result with an average accuracy of 98.4% and a detection speed of 47.6 FPS. It outperforms competing algorithms such as SSD, and Faster- RCNN, with a significant reduction in parameter cardinality and computational burden compared. In their research, Hu et al. [20] used the method of Optimal Gabor Filtering and Deep Feature Pyramid Network to utilize a military target detection dataset named MOD VOC, which was created to meet the PASCAL VOC dataset format standard and includes images primarily sourced from video footage captured by unmanned aerial vehicles (UAVs), ground cameras, and internet images. In doing so, they used five artificial intelligence algorithms (Faster R-CNN, DSOD300, DSSD513, YOLOv2 544, and their filtering method) to compare the individual performance of the best model for each algorithm. The best results for MOD VOC in terms of accuracy, recall, and average fps were achieved by their model in the amount of 88.76%, 78.45%, and 30.35 respectively, while the worst results were achieved by DSSD513 in the amount of 73.57%, 64.56% and 42.43. Kong et al. [21] used a military target dataset showing armed individuals with different weapons to improve the detection performance of the proposed YOLO-G algorithm. The authors introduced improvements compared to the YOLOv3 framework, including a lightweight GhostNet for improved accuracy and speed in detecting military targets. The dataset evaluation showed a 2.9% improvement in mAP and a 25.9 FPS increase in detection rate compared to the original YOLOv3, highlighting the effectiveness of their improved algorithm. Wang and Han [22] introduce the YOLO-M algorithm for military equipment target recognition, addressing challenges in small target detection. By incorporating the C3CMix module and modifying the activation function in YOLOv5, the proposed algorithm maintains high accuracy while reducing parameters, resulting in a 95.2% average accuracy, an 18.8% reduction in parameters, and a 14.5% decrease in computation. These improvements make YOLO-M well-suited for deployment in military equipment target recognition applications. Du et al. [23] investigated military vehicle object detection based on hierarchical feature representation and refined localization for the detection of military objects in the desert, grass, snow, city, and others. The authors applied R-FCNN, SSD, YOLOv3, YOLOv4, Faster R-CNN, and MVODM, i.e. a novelty algorithm created by the author. The models were trained on three different types of test data sets, i.e. large-scale, small-scale, and all subset test data sets. the best results were shown by MVODM (novelty algorithm) for a large-scale data set with evaluation metrics results in the amount of 85.6% mAP, while the worst performance was YOLOv3 for a small subset of data in the amount of 54.9% mAP. Nelson and McDonald [24] developed the Multisensor Towed Array System (MTADS), demonstrating its effectiveness in detecting buried unexploded ordnance with an outstanding probability of detection (0.95 or better). The system results highlight its precision in locating ordnance at self-penetrating depths, providing a cost-effective and accelerated approach to remediation compared to standard techniques. Pham and Polasek [19] address the challenges of surface object detection in urban environments, utilizing both infrared and visible spectra. The paper aims to develop an algorithm for detecting and selecting objects of interest, particularly civil automobiles resembling military equipment, captured by infrared and visible cameras in various outdoor conditions. The proposed algorithm involves determining optimal threshold values for image conversion in changing environmental conditions and colors, tested on static images, and extended to dynamic object detection, selection, and tracking through video processing. Additionally, the study emphasizes the use of threshold adjustment techniques to optimize object detection. Qin et al. [25] introduce a method inspired by EfficientDet trackers for classifying maritime military targets in high-resolution optical remote sensing images. The approach involves constructing a multilayer feature extraction network with attention mechanisms, utilizing ReLU activation, and employing deep feature fusion networks and prediction networks to accurately identify various types of military ships. The trained model was tested on six classes of vessels, achieving the best detection performance in the GW class with precision and recall values of 0.983 and 0.945, respectively. The lowest detection results were observed in the SS class, with precision and recall values of 0.974 and 0.822, respectively.” 

Additionally, in the methodology section, the following text has been added:

The YOLOv5 detection algorithm excels in achieving an optimal balance between speed and accuracy, featuring a refined architecture suitable for implementation on resource-constrained microcontrollers [30]. From the compact YOLOv5 nano model to larger variants, these models exhibit memory weights conducive to diverse applications, including military contexts. Comparative evaluations against DETR and EfficientDet underscore YOLOv5 superiority, particularly in challenges such as crop circle detection, where it outperforms with a recall of 0.98, surpassing DETR and EfficientDet 0.68 and 0.85, respectively, alongside precision values of 0.77 and 0.91 [30–32]. In scenarios involving overlapping object detection in kitchens, YOLOv5 excels by producing accurate frames and demonstrates superior performance compared to Faster R-CNN [33]. The study reveals YOLOv5’s accuracy of 0.8912 (89.12%) outshining Faster R-CNN’s 0.8392 (83.92%), underscoring its effectiveness in handling complex scenarios. Beyond object detection, YOLOv5 exhibits computational efficiency, outpacing sophisticated methods like RetinaNet, as observed [34–36]. Despite not universally achieving optimal results, the model’s performance nuances are vital considerations in addressing computational complexities during training.The versatility of YOLOv5 extends across domains, including the analysis of secondary waste treatment processes, applications in autonomous vehicles, and weed growth detection. Particularly noteworthy is its exceptional interference reduction and efficiency in weed growth detection, positioning YOLOv5 as an optimal choice for military applications, as emphasized [37]. Owing to foundational principles and analogous use cases, YOLOv5, characterized by its intricate, comprehensive, precise, and rapid attributes, undergoes scrutiny for the detection of military objects.

The given text is marked in red for one reason only, and that is to match the co-comment of Reviewer 1

Comment 3 

  1. It will be better to introduce some research questions addressed by your research.

Answer

The authors agree with the given comment, and hypothetical questions have been added to the introduction of this paper, i.e. the changes to the manuscript are as follows:

According to the presented literature overview and problem description, the following  questions can be asked: 

  • Is it possible to create a military data set by using publicly available data? 
  • Is it possible to use object detection algorithms such as YOLOv5 for military object  detection? 
  • How the proposed method simplifies and contributes to improving the quality of  military decision-making?

In the work, it is also necessary to adequately answer the posed hypothetical questions. With regard to the posed hypothetical questions, the answers are formulated in the introduction and conclusion of the paper, i.e. the manuscript. In the update of the new version of the paper, in the conclusion section, the given answers were further elaborated:

In conclusion, it is of utmost importance  to address the hypothetical questions posed at the outset of this work, namely:  

  • that it is possible through thorough research and study of multiple image materials to develop a sufficiently high-quality data set that will be used to train an artificial  intelligence model, or in this case, a detection algorithm, 
  • that it is possible to detect, classify, and localize objects such as flying objects, mobile  objects, etc. of military purpose by applying well-developed models such as YOLOv5,  and 
  • the completed methodology manifests qualitative results, with the application of which commanders of the armed forces can make decisions of considerable responsibility, eliminating occurrences of undesirable consequences. At the same time, taking  into account resources of lower performance on equipment that does not require high performance, decision-making is approached with optimal efficiency.

Comment 4 

  1. method is not novel. Please justify its importance for the dataset used.

Answer

The selection of YOLOv5 in the paper can be justified by several facts. YOLOv5 is known for its efficiency, speed, and accuracy, which makes it suitable for real-time object detection tasks. Its architecture has seen significant improvements over earlier versions, addressing limitations and improving performance. The decision to use YOLOv5 instead of more advanced frameworks such as DETR or other classical detectors such as Faster RCNN, SSD and RetinaNet can be significantly more complex, and further below are additional justifications that we consider YOLOv5 to be the optimal choice for this research work: 

Efficiency and Speed- YOLOv5 is renowned for its fast and efficient object detection capabilities, making it well-suited for scenarios where real-time processing is crucial. The study may prioritize speed and efficiency over some of the more complex but computationally demanding models like DETR.

Architectural improvement - YOLOv5 represents an evolution in the YOLO architecture, containing improvements and optimizations. These enhancements make it a competitive choice for certain applications, especially when considering a balance between accuracy and speed.

Task-specific considerations- The specific characteristics of the object detection task in the study align well with the strengths of YOLOv5. For instance, the dataset consists of numerous small objects and real-time performance is critical.

Simplicity of possibility for implementation- YOLOv5 is praised for its simplicity and ease of implementation, making it an attractive choice for researchers and practitioners who want a straightforward yet effective solution. Also, it opens a possibility for implementation on trivial microcontrollers such as Raspberry Pi.

The text in the "Methodology" section was also added, which further describes and emphasizes the importance of applying this algorithm in such situations. The amended text in the new version of the manuscript reads:

We opted to use YOLOv5 for the detection of military objects within a dataset we independently gathered. Our decision to utilize YOLOv5 over newer methods stemmed from the intention to showcase our capability to detect military objects within a publicly available dataset. YOLOv5 was chosen as it represents a stable and proven methodology for object detection tasks. Given its established track record and reliability, employing YOLOv5 allowed us to demonstrate our proficiency in identifying military objects reliably and accurately. While newer methodologies might offer enhancements or improvements, opting for YOLOv5 allowed us to focus on demonstrating our proficiency within a well-established and widely recognized framework. It ensured that our results could be easily reproducible and comparable against existing benchmarks in the field of object detection, reinforcing the reliability and stability of our approach.

Comment 5 

  1. Please give detailed analysis about the methods and parameter tuning along with proper descriptions about the dataset.

Answer

In the section 2.2.1 the following text was modified:”Each model was trained for 500 epochs with a single or stopping criterion of 100 epochs. The stop criterion plays a role in training cases, i.e. if no significant change occurs within 100 epochs, the training of the algorithm is stopped. Three adaptive moment estimation optimizers: Adam, AdamW, and stochastic gradient descent (SGD) were tested. ”, 

and it is replaced with: 

Each resulting model was trained over 500 epochs, using different image resolutions and optimizers. Starting with a resolution of 512 x 512 pixels, three models were trained, differing in the optimizer used - Adam, AdamW and the SGD optimizer. After these three models were trained, the training process became more complex, increasing the image resolution from 512 x 512 to 640 x 640. Likewise, these models were also trained with the aforementioned optimizers. Finally, the highest image resolution in this study was 1024 x 1024 pixels, with all three optimizers used for training for 500 epochs. The common element of all obtained models is the Patience parameter, which limits unnecessarily prolonged training. In other words, if the results do not improve significantly after 100 epochs, the training is stopped and the last obtained epoch is taken as the final training result.

Authors hope that the explanation and presentation in this version of the manuscript are clearer and improved compared to the previous version.

Comment 6 

  1. it will be better to compare the results of your approach with more recent methods in the literature.

Answer

The authors acknowledge the reviewer's comment, and the State-of-the-Art (SOTA) is now more comprehensive than in the last version of the manuscript. The current state-of-the-art is as follows:

Building upon the information at hand, and discerning the data source is achievable through the diverse resources mentioned earlier. Nonetheless, a pivotal inquiry arises concerning the identification of a spectrum of military installations. Bearing this in mind, numerous scholars have undertaken diverse research endeavors in this domain to delve more deeply into various methodologies for detecting adversary objects. Pham and Polasek  [19] investigated the development of a lightweight military target detection method, SMCA-YOLOv5. The method, which involves replacing the focusing module and redesigning the network structure, achieves an exceptional result with an average accuracy of 98.4% and a detection speed of 47.6 FPS. It outperforms competing algorithms such as SSD, and Faster- RCNN, with a significant reduction in parameter cardinality and computational burden compared. In their research, Hu et al. [20] used the method of Optimal Gabor Filtering and Deep Feature Pyramid Network to utilize a military target detection dataset named MOD VOC, which was created to meet the PASCAL VOC dataset format standard and includes images primarily sourced from video footage captured by unmanned aerial vehicles (UAVs), ground cameras, and internet images. In doing so, they used five artificial intelligence algorithms (Faster R-CNN, DSOD300, DSSD513, YOLOv2 544, and their filtering method) to compare the individual performance of the best model for each algorithm. The best results for MOD VOC in terms of accuracy, recall, and average fps were achieved by their model in the amount of 88.76%, 78.45%, and 30.35 respectively, while the worst results were achieved by DSSD513 in the amount of 73.57%, 64.56% and 42.43. Kong et al. [21] used a military target dataset showing armed individuals with different weapons to improve the detection performance of the proposed YOLO-G algorithm. The authors introduced improvements compared to the YOLOv3 framework, including a lightweight GhostNet for improved accuracy and speed in detecting military targets. The dataset evaluation showed a 2.9% improvement in mAP and a 25.9 FPS increase in detection rate compared to the original YOLOv3, highlighting the effectiveness of their improved algorithm. Wang and Han [22] introduce the YOLO-M algorithm for military equipment target recognition, addressing challenges in small target detection. By incorporating the C3CMix module and modifying the activation function in YOLOv5, the proposed algorithm maintains high accuracy while reducing parameters, resulting in a 95.2% average accuracy, an 18.8% reduction in parameters, and a 14.5% decrease in computation. These improvements make YOLO-M well-suited for deployment in military equipment target recognition applications. Du et al. [23] investigated military vehicle object detection based on hierarchical feature representation and refined localization for the detection of military objects in the desert, grass, snow, city, and others. The authors applied R-FCNN, SSD, YOLOv3, YOLOv4, Faster R-CNN, and MVODM, i.e. a novelty algorithm created by the author. The models were trained on three different types of test data sets, i.e. large-scale, small-scale, and all subset test data sets. the best results were shown by MVODM (novelty algorithm) for a large-scale data set with evaluation metrics results in the amount of 85.6% mAP, while the worst performance was YOLOv3 for a small subset of data in the amount of 54.9% mAP. Nelson and McDonald [24] developed the Multisensor Towed Array System (MTADS), demonstrating its effectiveness in detecting buried unexploded ordnance with an outstanding probability of detection (0.95 or better). The system results highlight its precision in locating ordnance at self-penetrating depths, providing a cost-effective and accelerated approach to remediation compared to standard techniques. Pham and Polasek [19] address the challenges of surface object detection in urban environments, utilizing both infrared and visible spectra. The paper aims to develop an algorithm for detecting and selecting objects of interest, particularly civil automobiles resembling military equipment, captured by infrared and visible cameras in various outdoor conditions. The proposed algorithm involves determining optimal threshold values for image conversion in changing environmental conditions and colors, tested on static images, and extended to dynamic object detection, selection, and tracking through video processing. Additionally, the study emphasizes the use of threshold adjustment techniques to optimize object detection. Qin et al. [25] introduce a method inspired by EfficientDet trackers for classifying maritime military targets in high-resolution optical remote sensing images. The approach involves constructing a multilayer feature extraction network with attention mechanisms, utilizing ReLU activation, and employing deep feature fusion networks and prediction networks to accurately identify various types of military ships. The trained model was tested on six classes of vessels, achieving the best detection performance in the GW class with precision and recall values of 0.983 and 0.945, respectively. The lowest detection results were observed in the SS class, with precision and recall values of 0.974 and 0.822, respectively.

Comment 7 

  1. it will be better to highlight the best results in the experimental results to indicate superiority of proposed approach.

Answer

The best results obtained from this research are now bolded in the modified version of the manuscript (Table 2.)

 Comment 8 

  1. It will be better to edit the manuscript with the native speaker to improve the quality of English.

Answer

The manuscript underwent a comprehensive review by a native speaker, during which any identified errors or logical inconsistencies were thoroughly addressed and resolved.

Reviewer 3 Report

Comments and Suggestions for Authors

1.      The collected dataset is quiet interesting. However there are problems with weapon included. You wrote that you classify military objects despite different forms of camouflage. Different countries have their own weapon camouflage and training a model to detect the military objects includes the problem to distinguish the country origin of weapon. Moreover camouflage sometimes helps to distinguish the kind of weapon. I think that this problem should include also research on the origin of weapon.

2.      People should also be detected on the video, because some of them may have weapon to destroy drone.

3.      There are no source of the drone, who captured the video. It is better to train the neural networks on the picture of resolution which is provided by real UAV like 4K UHD 2160p for DJI Mavic 3 or sth like that. The analysis of various UAVs image quality on various height would be useful in this article.

4.      One of the problem for UAS operator is also landscape and trees. They should also be considered to operate the drone in the correct way.

5.      Weight of neural  network should also be provided in the article because usually the UAS operator uses tablets and not pc.

6.      F1 Recall, Precision etc are quite famous formulas which are provided at each article on ML. There is no need to describe them.

7.      In my opinion, the most interesting part in this research is the collected dataset – so it should has more accurate description.

Author Response

Respected Reviewer,

We want to extend our sincere appreciation for your invaluable comments and insights. Your feedback has played a crucial role in improving the quality and clarity of our manuscript. We have taken careful note of all your suggestions and indicated the implemented changes in green throughout the manuscript. Your constructive input has greatly contributed to refining our work.

We thank you once again for dedicating your time to providing thoughtful feedback.

Best regards,

The Authors

Comments of the reviewer

Comment 1 

  1. The collected dataset is quiet interesting. However there are problems with weapon included. You wrote that you classify military objects despite different forms of camouflage. Different countries have their own weapon camouflage and training a model to detect the military objects includes the problem to distinguish the country origin of weapon. Moreover, camouflage sometimes helps to distinguish the kind of weapon. I think that this problem should include also research on the origin of weapon.

Answer

Dataset used in this manuscript is collected using open source (internet). First phase was collection of videos and photos, while second phase was cutting frames (from videos) and annotating videos and photos with annotation/labeling tool described in the manuscript. The dataset consists of various videos (different classes of weapon systems, different country of origin, different country’s versions, different angle of recording etc.) Some videos are actual combat footage (mostly from the current Ukraine situation) and some footage are from different country’s military exercises, expos and so on. Moreover some footage was recorded from air (mostly UAV footage) and some was recorded from ground. To conclude, the collected dataset consists of a variety of things (weapons in the open, weapons partly camouflaged, weapons in standby mode, weapons firing, aircrafts being shot down and so on). Therefore we think the current dataset partly represents things mentioned in the comment, however the research in which we would put more emphasis on camouflage is possible for future work.

In order to describe the method of data collection in more detail, the manuscript underwent additional modifications, which in the new version in subsection 2.1 read:

The dataset used in this work was obtained through open-source web sites and represents various military equipment and weapon systems used in modern combat.  Those weapon systems have different countries of origin and were recorded from different angles and in different conditions. Some weapon systems were recorded from the air by UAV, while others were recorded from the ground by humans. Moreover, the dataset has examples of maneuver forces' vehicles such as tanks, infantry fighting vehicles, and armored personnel carriers, examples of air force vehicles such as transport or assault helicopters and transport or assault airplanes, and examples of engineering vehicles, anti-aircraft vehicles, and artillery vehicles. This research’s focus is weapon systems only, therefore people were not included. Furthermore, a large portion of collected videos are real combat footage, recorded mostly in the current conflict between the Russian Federation and Ukraine. Other videos were taken at various military exercises or expos and posted on the internet.

Comment 2 

  1. People should also be detected on the video, because some of them may have weapon to destroy drone.

Answer

The main idea of this research was to detect, classify and locate only military equipment. However, people (soldiers or civilians) are also an important element of modern combat and we might include them in future work. According to comment following sentence is added into new version of manuscript (lines 566-568):

Moreover, including humans (soldiers or civilians) into one or more classes would be highly beneficial because of their importance in modern combat.”

Comment 3 

3.There are no source of the drone, who captured the video. It is better to train the neural networks on the picture of resolution which is provided by real UAV like 4K UHD 2160p for DJI Mavic 3 or sth like that. The analysis of various UAVs image quality on various height would be useful in this article.

Answer

Collected footage was not only obtained by UAV, therefore it is not UAV only problem. Dataset includes videos from all angles and perspectives. Moreover, most of the dataset is combat footage. In our opinion it is hard to imagine, all armies would have access to 4K resolution cameras, UAVs that would be able to carry that kind of payload and would be able to carry out real missions (at least 60 km from your controlling station. In our opinion most armies (operating UAS) have access to 480p or 720p resolution. However this is a good point for future work and we will definitely take it into consideration. 

In the revised version of the manuscript in the conclusion section the following changes are made (line 568-572): 

Finally, it is of utmost importance to implement the model and assess its performance in real-world scenarios. The evaluation can be conducted using videos obtained by commercial UAVs equipped with high-resolution payloads, such as 4K UHD 2160p. However, optimal evaluation would involve the use of a military-grade payload due to its relevance in real-world applications.

Comment 4 

4.One of the problem for UAS operator is also landscape and trees. They should also be considered to operate the drone in the correct way.

Answer

According to this comment, the following paragraph is added to introduction:

“The aim of this article is not to showcase the methodology of UAV control, but rather to demonstrate how a dataset with military applications can be compiled using publicly available data, which can be utilized for the development of detection software. For this reason, the paper does not focus on the construction elements of drones or the methodology of their flight control.”

Comment 5 

  1. Weight of neural  network should also be provided in the article because usually the UAS operator uses tablets and not pc.

Answer

YOLOv5 nano small models are much less demanding than medium large or extra large models and are quite suitable for implementation on computer-poor devices such as Raspberry pi microcontrollers, which is shown in the new version of the manuscript. The only difference between the use of a personal computer and a tablet would be its performance, that is, the inference time and possibly the fps from the camera, which is actually shown in Table 5 in milliseconds. In the new version of the manuscript, Table 6 was added, which was introduced at your request, and shows the memory requirement in megabytes for each obtained model.

Also at the end of section 3.2, the text describing Table 6 was added, namely: 

"The significance of employing the YOLOv5 algorithm for military object detection is underscored by the insights provided in Table 6. The weight of the obtained YOLOv5 model, ranging from 13 to 14 megabytes, is remarkably compact, presenting opportunities for deployment on resource-constrained devices, such as Raspberry Pi [47 - 49].

Optimizer

Resolution [ Height X Width]

Size [MB] 

Adam

512 X 512

13.6

Adam

640 x 640

13.7

Adam

1024 x 1024

13.8

AdamW

512 X 512

13.6

AdamW

640 x 640

13.7

AdamW

1024 x 1024

13.8

SGD

512 X 512

13.6

SGD

640 x 640

13.7

SGD

1024 x 1024

13.8

The reviewer’s claim is actually confirmed, but not from the stand of memory capacity, but from a performance standing, with a sentence in the new version of the manuscript:

 "However, it is crucial to note that for efficient data processing and inference, a moderately powerful graphics unit, particularly one containing a GPU, is still mandatory. "

Also, it is worth mentioning that in real combat conditions, UAV crews (consisting of operators, technicians, and CIS (communication information systems) technicians have PCs present on site. They have it because they need to have constant communication with their headquarters to keep constant command and control. PCs are also present to stream live video from UAVs to decision-makers. 

Comment 6 

  1. F1 Recall, Precision etc are quite famous formulas which are provided at each article on ML. There is no need to describe them.

Answer

The authors agree with the reviewer's comment, and the entire sub-section 3.2 has been removed. The new version of the manuscript no longer includes the sub-section on “3.2 Evaluation metrics.”

Comment 7  

  1. In my opinion, the most interesting part in this research is the collected dataset – so it should has more accurate description.

Answer

According to the comment dataset is described with a more accurate description in following paragraph:

 “ The dataset used in this work was obtained through open-source web sites and represents various military equipment and weapon systems used in modern combat. Those weapon systems have different countries of origin and were recorded from different angles and in different conditions. Some weapon systems were recorded from the air by UAV, while others were recorded from the ground by humans. Moreover, the dataset has examples of maneuver forces' vehicles such as tanks, infantry fighting vehicles, and armored personnel carriers, examples of air force vehicles such as transport or assault helicopters and transport or assault airplanes, and examples of engineering vehicles, anti-aircraft vehicles, and artillery vehicles. This research’s focus is weapon systems only, therefore people were not included. Furthermore, a large portion of collected videos are real combat footage, recorded mostly in the current conflict between the Russian Federation and Ukraine. Other videos were taken at various military exercises or expos and posted on the internet.

Round 2

Reviewer 1 Report

Comments and Suggestions for Authors

The manuscript has significantly improved after the 1st round of review. However, the contribution of this work is marginal and is not clearly highlighted in the abstract and introduction. In my opinion, the main contribution of this work should be the curation of a new dataset for military weapon detection rather than any invention or refinement of detection algorithms. The authors do not propose new modules/revisions to the existing YOLO framework but simply apply it to this new dataset and report the results. From this perspective, this can be a good technical report but not an academic paper. That being said, I encourage the authors to highlight the contributions of a new dataset and to justify why it is necessary to have such a new dataset. 

Comments on the Quality of English Language

The presentation needs to be improved.

Author Response

Respected Reviewer,

In response to your feedback, we have made additional modifications to the manuscript. An extra subsection has been included, offering a more detailed account of the dataset collection and emphasizing its importance. Additionally, within this sub-subsection, we have added content addressing the significance of the data set both from the perspective of Machine Learning and the military operator.

In section 2.1 Materials we have added following sentences:

The dataset used in this work was obtained through open-source websites. The websites included were various official countries’ armies’ websites, different country’s official army’s profiles on popular video streaming platforms, open-access gore websites, and news portals. The used dataset represents various military equipment and weapon systems used in modern combat.”

Following subsection: 2.1 "Material," we have undertaken additional modifications to the manuscript. Subsequently, we have introduced sub-subsection 2.1.1, the contents of which are now articulated as follows:"

2.1.1 Military data curation process: Unveiling the significance and challenges of comprehensive dataset in a strategic context

The quality of the dataset is a key prerequisite for successful research in the field of artificial intelligence, and this especially applies to the development and training of algorithms. According to generally accepted standards, as much as 70 % of the total effort in the process of training an AI algorithm is devoted to the collection, processing, and preparation of data. In order for research to achieve an acceptable level of precision and reliability, it is essential to have a high-quality dataset.

Figure 3 illustrates the arduous nature of the dataset collection process, comprising a series of sub-steps that demand considerable time investment. The compilation of a dataset to a level acceptable for annotation, let alone training AI algorithms, is a painstaking endeavor. This visual representation underscores the pivotal role of a well-curated dataset in the development of artificial intelligence. It highlights that research based on vaguely defined or low-quality data may yield unrealistic results and draw incorrect conclusions. Various methods, including surveys, camera recordings across different devices, and even oral transmission of information, can be employed in data collection. Diverse collection conditions, encompassing factors such as the environment, weather conditions, and context, can exert a substantial impact on research outcomes. Hence, it becomes crucial to meticulously account for these factors when curating a dataset.

Figure 3. Flowchart utilized in this research for military data collection protocol

As depicted in Figure 3 the data acquisition process comprises two pivotal phases. The initial phase entails a comprehensive exploration of available videos and diverse image repositories accessible on the internet. The primary objective of this stage was to meticulously curate a set of high-caliber computer data that would undergo subsequent processing. The quest for data traversed various platforms, including YouTube, Google Images, Wikimedia Commons, and others.

Prior to commencing the analytical phase, meticulous consideration was devoted to adapting and scrutinizing all designated video formats. This involved the critical assessment of video quality, addressing challenges related to perspective constraints, accounting for temporal variables, and discerning potential manipulations and edits within the material. Subsequent to the successful compilation of a substantial dataset, comprising 101 videos averaging 180 seconds in length, each video comprising 60 frames per second (FPS) for a cumulative total of 1090800 images, a judicious analysis and evaluation process ensued. Each image underwent scrutiny as a prospective candidate for annotation, necessitating precise labeling and identification of pertinent information to ensure a nuanced and pertinent analysis in subsequent research endeavors. Following the implementation of solutions to potential challenges, a comprehensive review process was undertaken to assess all acquired frames. The primary objective was to ascertain the presence of objects exhibiting military characteristics, thereby enhancing the overall quality of the dataset and refining the definition of the target class of military objects. Should a specific frame fail to meet the pre-established criteria, signifying a departure from the defined conditions, it was systematically rejected from further consideration. This meticulous curation process ensured that only frames aligning with the specified criteria were retained, contributing to the precision and reliability of subsequent analyses and findings within the research context. Following the successful resolution of potential challenges, a meticulous examination of all acquired frames was initiated, with the objective of discerning objects exhibiting military characteristics. The primary objective of this procedure was to enhance the quality of the dataset and provide a more accurate delineation of the presence of the targeted class of military objects. Instances where the specified conditions were not met, where a particular frame failed to meet the predefined criteria prompted its systematic exclusion. This systematic curation process was undertaken to uphold the consistency and high quality of the data utilized in the course of the research.

Upon the successful extraction and meticulous curation of images, the subsequent step involved annotating the images to facilitate the training of the YOLOv5 detection algorithm. In this phase, dual considerations were paramount. Firstly, the military perspective was taken into account, encompassing elements deemed significant from the standpoint of a soldier, lieutenant, and the like. Simultaneously, the YOLOv5 algorithm was loaded and subjected to rigorous testing to evaluate its performance under real-world conditions. A more intricate exposition of the data collection process, along with illustrative examples, is presented in Figures 4 – 9.

From military operations planners’ and military commanders’ standpoint, the significance of this dataset lies in the variety of military equipment classes included. An algorithm trained with this particular dataset can help military commanders and military headquarters to have better situational awareness about opposition forces’ structure and operational capabilities. Especially in intense battle rhythm operations, when quick information flow is essential. From a machine learning standpoint, the significance of this dataset lies in its pivotal role in enhancing the robustness and generalization of algorithms. The incorporation of luminance contrast diversifies the learning experience, enabling the algorithm to adeptly discern varying levels of luminance and thus bolstering its resilience to changes in lighting conditions. Furthermore, the dataset richness in colors and textures contributes to the capacity of the algorithm to generalize across diverse object types and backgrounds, as exemplified in Figures 7 and 8. In terms of preventing overfitting, the dataset inclusion of different perspectives is instrumental in averting model specialization to particular positions or viewing angles. Moreover, the incorporation of varied recording conditions, such as distinct cameras and weather scenarios, serves as a safeguard against overfitting to a specific dataset, as demonstrated in Figures 5 and 7. The dataset emphasis on increasing variation is evident through its incorporation of geographical and environmental diversity. This inclusion exposes the algorithm to different locations and environments, fostering an ability to adapt to various conditions, such as multiple entry with similar properties as shown on Figure 9. Notably, the dataset captures scenarios where autonomous humans (AH) are situated outdoors or within dense vegetation, such as tall trees or the elevated roofs of buildings. The dataset further contributes to the model's versatility in solving various problems. The introduction of different object sizes and distances enables the model to develop proficiency in accurately detecting objects within diverse contexts. This is illustrated, for instance, in Figure 4, where AHs exhibit scaling, with one AH appearing smaller in relation to the other. Additionally, the dataset encompasses variations in object positions within images, facilitating the model's ability to recognize objects across different parts of an image. Lastly, the inclusion of diverse time periods in the dataset, encompassing night, day, snow, dust, smoke and more, augments the model's performance on real-world data as on Figure 6. This is particularly relevant in the context of autonomous vehicles, where variations in driving conditions, including night, day, rain, and snow, contribute to preparing the model for a spectrum of real road situations.

As perceived from the perspective of ML engineers and AI algorithms, the presented dataset furnishes exemplary instances under diverse conditions. This dataset serves as a comprehensive evaluation ground, testing the algorithm’s performance capabilities in varied scenarios. Additionally, it caters to the specific needs of military operators, providing valuable insights tailored to their operational requirements.

"

All the mentioned changes in the manuscript are highlighted in red, and the authors of this paper hope that the manuscript has now reached an acceptable level for publication.

Sincerely,

The Authors

Reviewer 2 Report

Comments and Suggestions for Authors

The authors have addressed all of my concerns. It looks ready for the publication.

Comments on the Quality of English Language

Minor English edit is required.

Author Response

Respected Reviewer,

We want to express our appreciation for the helpful comments you shared during the review. Your feedback played a key role in shaping the modifications to the latest version of the manuscript. We value your time and insights, and we're grateful for your contribution to improving our work.

Thanks again for your feedback.

The authors

Round 3

Reviewer 1 Report

Comments and Suggestions for Authors

No further comments.